# Adaptive Acquisition Selection for Bayesian Optimization with Large Language Models

**Giang Ngo,**\* **Dat Phan-Trong, Dang Nguyen, Sunil Gupta & Svetha Venkatesh**
Applied Artificial Intelligence Initiative, Deakin University
Victoria, Australia
{g.ngo,d.phantrong,d.nguyen,sunil.gupta,svetha.venkatesh}@deakin.edu.au

## Abstract

Bayesian Optimization critically depends on the choice of acquisition function, but no single strategy is universally optimal; the best choice is non-stationary and problem-dependent. Existing adaptive portfolio methods often base their decisions on past function values while ignoring richer information like remaining budget or surrogate model characteristics. To address this, we introduce LMABO, a novel framework that casts a pre-trained Large Language Model (LLM) as a zero-shot, online strategist for the BO process. At each iteration, LMABO uses a structured state representation to prompt the LLM to select the most suitable acquisition function from a diverse portfolio. In an evaluation across 50 benchmark problems, LMABO demonstrates a significant performance improvement over strong static, adaptive portfolio, and other LLM-based baselines. We show that the LLM's behavior is a comprehensive strategy that adapts to real-time progress, proving its advantage stems from its ability to process and synthesize the complete optimization state into an effective, adaptive policy.

## 1 Introduction

Bayesian Optimization (BO) is a preeminent methodology for the global optimization of expensive-to-evaluate, black-box functions, a pervasive challenge across science and engineering (Shahriari et al., 2015). Its framework uses a surrogate model (often a Gaussian Process (GP)) to approximate the objective function and an acquisition function (AF) to intelligently select the next point to evaluate, balancing the trade-off between exploration and exploitation. A core challenge in BO is the selection of the AF. It is well-established that no single, fixed AF offers optimal performance across all problems (Hoffman et al., 2011); the best strategy is highly contingent on the characteristics of the objective function and can even change dynamically throughout a single optimization run. This has spurred the development of adaptive strategies that move beyond static AF selection to dynamically choose different AFs from a portfolio every iteration (Hoffman et al., 2011).

Existing adaptive portfolio methods, however, suffer from a critical limitation: their decisions are guided almost exclusively by a narrow view of the optimization process, typically relying on the history of observed function values. Portfolio-based strategies (Hoffman et al., 2011; Vasconcelos et al., 2019; 2022) operate on a reward signal derived from the past performance of each AF, utilizing only the function evaluations and the surrogate model's output. These methods largely ignore a wealth of other critical state information, such as the remaining optimization budget, the distance between evaluated points, and insights about the function's complexity encoded in the surrogate model's own hyperparameters (e.g., GP lengthscales). The core challenge is that designing a principled, algorithmic approach that can effectively reason over such a diverse and complex set of strategic, tactical, and landscape-related information has proven immensely difficult.

This paper bridges this gap by leveraging a Large Language Model (LLM) as a dynamic optimization strategist. We are motivated by the fact that modern LLMs, trained on immense corpora of scientific literature and code, possess a rich, nuanced, and implicitly encoded understanding of optimization principles. Instead of hand-crafting a complex policy covering all state information, we tap into this

---

\*corresponding author

pre-trained knowledge, as well as the reasoning capabilities of the LLM to guide the exploration-exploitation balance in an optimization process. We introduce a new formulation of adaptive BO: casting acquisition function selection as an *in-context decision-making problem* solved by a pre-trained LLM, supported by a novel state serialization design. At each iteration, the complete, multi-faceted optimization state is serialized into a structured textual prompt. The LLM then analyzes this rich state and select the most appropriate AF from a diverse portfolio for the immediate next step.

Experiments show LMABO consistently outperforms the baselines across diverse optimization problems. Ablation studies confirm the LLM leverages all components of the state summary, with performance dropping when any part is removed. A behavior analysis reveals a comprehensive strategy: LMABO prefers exploration when progress has stagnated, heavily exploits with low remaining budget, and aggresively switches between AFs; during early stages of optimization, the LLM is sensitive to all information; during middle stages, the performance history and process status are influential and towards the end, sensitive to only incumbent values. This proves that LMABO's success stems from its ability to reason over a complete set of strategic and tactical information and to successfully *mirror the established best practices in BO* similar to the intuition of a human expert.

The contributions of this work can be summarized as follows:

- We recast BO's acquisition function selection as an in-context decision-making task with an LLM as a closed-loop strategist to select the most appropriate AF.

- We propose a structured representation of the BO state, shown via ablations to be essential for effective zero-shot decision-making.

- On 50 benchmarks, LMABO outperforms static, adaptive, and LLM-based baselines, with analysis revealing emergent, state-aware strategies beyond simple heuristics.

## 2 BACKGROUND

**Bayesian Optimization** is a sample-efficient framework for optimizing expensive-to-evaluate black-box functions (Frazier, 2018). Formally, let $f : \mathcal{X} \to \mathbb{R}$ be an unknown objective function defined over $\mathcal{X} \subset \mathbb{R}^d$. BO constructs a probabilistic surrogate model, typically a Gaussian Process (Rasmussen & Williams, 2005), to approximate $f$ and quantify uncertainty in unexplored regions. At each iteration, given the set of previously observed evaluations $\mathcal{D}_{t-1}$, BO selects the next query point $x_t \in \mathcal{X}$ by maximizing an *acquisition function* $\alpha(x; \mathcal{D}_{t-1})$, which balances *exploration* (sampling in uncertain regions) and *exploitation* (sampling near low predicted values (for minimization)). There are many well-studied and effective AFs, including Expected Improvement (EI) (Mockus, 1998), Thompson Sampling (TS) (Chowdhury & Gopalan, 2017), and Upper Confidence Bound (UCB) (Srinivas et al., 2010), and each of them has their own advantages and disadvantages. Some AFs like TS and UCB (given appropriate hyperparameters) are more explorative, while others like EI are more exploitative. However, no single AF performs optimally across all problems (Hoffman et al., 2011), and achieving sample efficiency often requires dynamically prioritizing exploration or exploitation at the appropriate stages of the optimization process. Therefore, using the right AF with the right focus on exploration or exploitation at the right time can be a key to better BO performance. More details about AFs used in this paper and their grouping are provided in Appendix A.

A **Gaussian Process** is a nonparametric prior over functions, defined such that any finite collection of function values follows a joint Gaussian distribution. A GP is fully specified by its mean function $m(x)$ and covariance kernel $k(x, x')$, where the kernel encodes assumptions about the smoothness and structure of the underlying function. An example of a kernel is the squared exponential kernel:

$$k(x, x') = \sigma_f^2 \exp\left(-\tfrac{1}{2} \sum_{i=1}^{d} \frac{(x_i - x_i')^2}{\ell_i^2}\right),$$

where $\ell_i$ are the *lengthscales*, controlling function variation along each input dimension, and $\sigma_f^2$ is the *outputscale*, determining the overall variance of the function values. A small lengthscale means the function varies rapidly with input changes, and a small outputscale keeps predictions close to the mean with little variation. These hyperparameters are typically learned by maximizing the marginal likelihood of observed data, enabling GPs to adapt their complexity to the underlying objective.

## 3 RELATED WORKS

**Portfolio-Based Strategies**   These methods treated AF selection as a portfolio allocation problem. GP-Hedge (Hoffman et al., 2011) framed the task as a *multi-armed bandit* problem, selecting AFs by first computing a reward signal derived from their past performance then randomly sampling an AF according to a probability distribution weighted by these rewards. Subsequent methods like No-PASt-BO (Vasconcelos et al., 2019) and SETUP-BO (Vasconcelos et al., 2022) sought to improve upon this by introducing memory factors to discount distant evaluations, but they still rely on the same fundamental reward mechanism. Shahriari et al. (2014) used an information-theoretic approach with Entropy Search Portfolio (ESP), which shifted the selection criterion to a forward-looking measure of utility: the expected reduction in uncertainty about the location of the global optimum. However, a key limitation unites these strategies: their decisions are guided by a narrow view of the optimization state, focusing primarily on function values and uncertainty while ignoring other critical information like the remaining budget or characteristics of the surrogate model itself.

**Learning-Based Strategies**   MetaBO (Volpp et al., 2020) and FSAF (Hsieh et al., 2021) meta-learn a state-dependent policy, formalizing AF selection as a reinforcement learning problem. These are related but not applicable to our setting, as they are designed for a transfer learning scenario, where a policy is learned on a distribution of source tasks and adapted to a new target task.

**LLM-Based Bayesian Optimization**   With the recent success of LLMs, efforts have emerged to incorporate them into the BO process, though their functional roles differ fundamentally from our approach. FunBO (Aglietti et al., 2025) uses the LLM as an *offline algorithm generator* to discover new AFs with a few function evaluations, but it does not participate in the adaptive process of AF selection. Other recent methods have integrated LLMs more directly into the optimization loop as *component replacements*. For instance, LLMP (Requeima et al., 2024) focuses on enhancing the surrogate model with natural language priors, while LLAMBO (Liu et al., 2024) uses the LLM for multiple steps in the optimization process including generating initial samples, surrogate modeling, and proposing candidate points. These approaches rely on the LLM's inductive biases regarding numerical function shapes, effectively treating the text-based model as a numerical regression engine. LMABO establishes a distinct paradigm: the **semantic controller**. Unlike FunBO, LMABO operates online to adapt to real-time feedback. Unlike LLAMBO/LLMP, it does not replace the rigorous mathematical backbone (Gaussian Processes) with LLM predictions. Instead, it recasts the AF selection itself as a sequential decision-making task solvable by in-context reasoning, representing a new paradigm for adaptive BO.

## 4 METHODOLOGY

### 4.1 LANGUAGE MODEL-ASSISTED ADAPTIVE BAYESIAN OPTIMIZATION (LMABO)

Our proposed method LMABO uses a pre-trained LLM to dynamically select the most appropriate acquisition function at each iteration of the BO process. The framework operates as a closed-loop system, where the LLM is prompted at each iteration with a rich representation of the optimization state to infer the most effective AF for evaluation. The entire process is detailed in Algorithm 1.

### 4.2 THE LLM AS A ZERO-SHOT STRATEGIST

A core tenet of our LMABO framework is to leverage the reasoning capabilities of a large, pre-trained LLM in a zero-shot setting. This approach requires no task-specific fine-tuning to the LLM's weights. Instead, the model's strategic behavior is guided entirely through in-context learning, conditioned on an initial prompt, $P_0$, that structures the entire decision-making task. The initial prompt, $P_0$, is a static instruction set that establishes the context for the entire optimization run. It is composed of several key components designed to elicit an expert-like decision-making process:

1. **Role-playing Instruction:** The prompt begins by providing the LLM with an instruction to act as an "expert in Bayesian Optimization". This contextual framing is used to condition the model, leveraging the patterns it learned during pre-training to emulate the decision-making process that a human expert might follow when presented with similar data.

---

**Algorithm 1** The LMABO Framework

---

**Require:** Objective function $f(\mathbf{x})$; Initial dataset $\mathcal{D}_0 = \{(\mathbf{x}_0, y_0), \dots\}$; Optimization budget $T$; Portfolio of acquisition functions $\mathcal{A} = \{\alpha_1, \alpha_2, \dots, \alpha_K\}$; Large Language Model $\Psi$.

1: Construct an initial prompt $P_0$ that defines the LLM's role as a BO expert and provides the set of available acquisition functions $\mathcal{A}$. ▷ See Appendix C
2: Send $P_0$ to $\Psi$ to establish context. ▷ See Sec. 4.2
3: **for** $t = 1, 2, \dots, T$ **do**
4:     Fit a Gaussian Process model $\mathcal{GP}_{t-1}$.
5:     Generate state summary $S_t$ from $\mathcal{GP}_{t-1}$ and optimization history. ▷ See Sec. 4.3
6:     Construct the update prompt $P_t$ from $S_t$.
7:     Obtain the next acquisition function $\alpha_t \leftarrow \Psi(P_t)$. ▷ See Sec. 4.2
8:     Optimize $\alpha_t$ to find the next point to evaluate: $x_t \leftarrow \arg\max_x \alpha_t(x)$.
9:     Evaluate the true objective function: $y_t = f(x_t) + \eta_t$ with noise $\eta_t$.
10:     $\mathcal{D}_t \leftarrow \mathcal{D}_{t-1} \cup \{(x_t, y_t)\}$.
11: **end for**
12: **return** The point $x^*$ corresponding to the best function value in $\mathcal{D}_T$.

---

2. **Available Actions:** We explicitly define the portfolio of available acquisition functions, $\mathcal{A}$. Each function is listed with its abbreviation (e.g., EI, UCB) and full name. We refrain from giving a description for each AF to avoid biased interpretations and instead rely on the LLM's encoded knowledge. Note that we default to UCB if the LLM's output is invalid.

3. **State Information Schema:** The prompt describes the structure of the state summaries, $S_t$, that it will receive at each subsequent step, explaining what each piece of information (e.g., GP lengthscales, current performance) represents.

4. **Output Formatting Constraint:** Finally, the prompt specifies a strict output format ("Acquisition abbreviation: Justification") to ensure responses can be reliably parsed.

$P_0$ is sent once at the beginning of the optimization to set the stage. At each iteration $t$, the update prompt $P_t$ is formed by appending the new state summary, $S_t$ (see the next section), to the established context of $P_0$. The full text for $P_0$ is provided in Appendix C for reproducibility.

### 4.3 OPTIMIZATION STATE REPRESENTATION

A key component of our LMABO framework is the translation of the high-dimensional, numerical state of the BO process into a concise, human-readable textual summary, $S_t$. This summary is designed to provide the LLM with a comprehensive, multi-faceted view of the optimization landscape and progress. The state summary $S_t$ at each iteration $t$ is composed of the following elements:

- **Process Status:** To contextualize the current step within the overall process, we provide the number of **evaluations performed** so far ($N$), the **remaining budget** ($N_{\text{rem}}$), and the problem **dimensionality** ($D$). The remaining budget, in particular, is critical for balancing the long-term need for exploration against the short-term need for exploitation.

- **Performance History:** To provide context on the optimization's progress, we include several performance indicators. These are the **incumbent objective value** ($f_{\text{min}}$), the observed **function value range**, and the **shortest distance** from the last evaluated point to any previous point (as an indicator of the last evaluation's exploration tendency).

- **GP Model Characteristics:** To inform the LLM about the current understanding of the function landscape, we provide key hyperparameters from the fitted surrogate model $\mathcal{GP}_{t-1}$. This includes the kernel's **outputscale** and summary statistics of the **lengthscales** (minimum, maximum, mean, and standard deviation).

These components are formatted into a structured string that becomes the core of the prompt $P_t$ sent to the LLM at each iteration. See examples of these state summaries in Appendix C. The design of $S_t$ balances compactness with completeness, enabling the LLM to leverage domain-specific signals (budget, GP hyperparameters, exploration metrics) without requiring training. Our ablation results (Table 2) demonstrate that omitting any of these elements significantly degrades performance, underscoring the importance of the representation.

## 5 EXPERIMENTS

### 5.1 EXPERIMENT SETUP

**Baselines** We employ a comprehensive set of baselines spanning four categories:

- **Static Acquisition Functions:** These are standard and popular BO methods that use a single, fixed AF throughout the process. We include all 12 AFs that constitute the portfolio from which LMABO can select (see Appendix A for details).

- **Simple Meta-strategies:** These methods use simple, non-adaptive strategies to select among multiple AFs. They first include the naive strategy of uniformly random selection within a portfolio of: 1) all 12 AFs, 2) most popular AFs in practice (EI, TS, UCB, PosMean), and 3) AFs that are commonly selected by LMABO (i.e. EI, LogEI, and TS, as shown later in Figure 2a). We also include strategies that alternate between exploration (TS) and exploitation (EI) (i.e. Alt-EI-TS-$k$ with $k = 1, 3, 5$) and a strategy that explores first then exploits (i.e. TwoPhases-TS-EI).

- **Adaptive Acquisition Functions:** These methods adapt their AF based on the optimization state, including GP-Hedge (Hoffman et al., 2011), No-PASt-BO (Vasconcelos et al., 2019), SETUP-BO (Vasconcelos et al., 2022), and ESP (Shahriari et al., 2014).

- **LLM-based Methods:** State-of-the-art baselines that incorporate LLMs into the BO process, including LLAMBO (Liu et al., 2024) and LLMP (Requeima et al., 2024).

**Benchmark Problems** The evaluation is performed on a broad set of 50 problems to test for robustness and general applicability. These includes 30 synthetic benchmark functions from the COCO platform (Hansen et al., 2021) and the BoTorch library (Balandat et al., 2020). In addition, we use 20 real-world hyperparameter optimization problems from Bayesmark (Uber, 2020). This benchmark evaluates the practical applicability of LMABO on a common and important task in machine learning. Details of these benchmark problems are provided in Appendix B.

**Implementation Details** We implement LMABO with Gemini-2.5 Flash. For surrogate modeling of GP-based methods, we use a GP with a Matérn 5/2 kernel, and the implementations are built using standard modules from the BoTorch library. Each optimization run is initialized with $2D + 1$ points, where $D$ is the dimensionality of the problem. The optimization budget is set to 50 iterations for problems with fewer than 10 dimensions and 100 iterations for problems with 10 or more dimensions. More implementation details are provided in Appendix B.

**Evaluation** Each experiment is repeated 10 times with different random seeds. For each method on each problem, we averaged the **Areas Under the Simple Regret Curves** (AUCs) over all 10 repetitions. We then compute **Relative Performance** (RP): for each problem, the method with the lowest (best) total AUC receives an RP of 1.0, and all other methods are assigned an RP equal to their total AUC divided by the best total AUC for that problem. This ensures aggregation across problems is not affected by different absolute scales of AUCs. A **rank** of each method on each problem is determined by sorting all methods by their total AUC (lower is better), assigning rank 1 to the best. Note that the ranking includes 25 baselines, 8 ablation variants of LMABO, and 4 adaptive portfolio variants, resulting in a maximum rank of 38. RP and rank help provide a clear and concise summary of comparative performance across all methods and problems instead of plotting all regret curves for 38 methods on 50 problems. Experimental results will undergo a rigorous statistical analysis to ensure the validity of our findings. Details about the statistical tests are provided in Appendix B.3.

### 5.2 MAIN RESULTS

The results, summarized in Table 1, demonstrate that LMABO achieves a substantial performance improvement over all baseline categories. Averaged across problems, LMABO's total AUC is **9.7%** lower than the best static AF, **14.8%** lower than the best simple meta-strategies, **54.7%** lower than the best LLM-based method, and **16.6%** lower than the best adaptive portfolio method; consequently, LMABO ranks among the top four methods on average. LMABO's low variation (CV = 0.37) indicates high consistency across different seeds. A 50-iteration run with LMABO uses a total

Table 1: **Overall performance comparison of LMABO against all baselines across 50 optimization problems**. P-values from Friedman tests in the last row indicate statistically significant differences among methods for both RP and rank. The third and fifth columns show p-values of post-hoc pairwise comparisons between LMABO and each method, which confirm that the differences in performance between LMABO and all methods are significant. Exploitative AFs are marked in blue and explorative AFs are marked in magenta (see Appendix A for details).

| Method | Mean RP (Interquartile Range) | P-value (RP) | Mean Rank (Min - Max) | P-value (Rank) | CV of AUC |
|---|---|---|---|---|---|
| *Static Acquisition Functions* | | | | | |
| PI | 1.53 (1.08–1.53) | 3.71e-03 | 15.80 (1–36) | 1.54e-04 | 0.45 |
| LogPI | 1.40 (1.10–1.47) | 7.40e-03 | 14.52 (1–37) | 3.89e-04 | 0.47 |
| EI | 1.34 (1.11–1.52) | 2.85e-04 | 13.08 (1–34) | 3.87e-05 | 0.44 |
| LogEI | 1.36 (1.15–1.48) | 8.24e-06 | 13.92 (1–35) | 1.49e-06 | 0.42 |
| PosMean | 1.42 (1.08–1.51) | 1.05e-02 | 14.70 (1–37) | 5.06e-04 | 0.45 |
| PosSTD | 7.02 (2.12–5.51) | 3.37e-08 | 34.78 (3–38) | 3.47e-08 | 0.48 |
| UCB | 1.75 (1.23–2.02) | 2.59e-07 | 23.94 (1–37) | 1.32e-07 | 0.37 |
| TS | 2.07 (1.32–1.92) | 1.10e-07 | 25.46 (1–37) | 7.05e-08 | 0.35 |
| KG | 1.66 (1.24–1.78) | 7.19e-08 | 23.96 (4–36) | 5.09e-08 | 0.40 |
| PES | 2.93 (1.68–3.24) | 2.80e-08 | 31.92 (10–38) | 2.74e-08 | 0.38 |
| MES | 2.80 (1.22–1.66) | 5.16e-07 | 20.54 (1–37) | 2.04e-07 | 0.40 |
| JES | 1.62 (1.30–1.70) | 4.30e-08 | 24.64 (1–36) | 4.79e-08 | 0.39 |
| *Simple Meta-strategies* | | | | | |
| Random (Full portfolio) | 1.43 (1.23–1.54) | 5.16e-07 | 17.36 (2–31) | 1.91e-07 | 0.41 |
| Random (EI, TS, UCB, PosMean) | 1.45 (1.19–1.55) | 1.82e-07 | 17.62 (1–34) | 2.07e-07 | 0.42 |
| Random (EI, LogEI, TS) | 1.42 (1.18–1.54) | 7.91e-06 | 15.62 (1–33) | 3.60e-06 | 0.40 |
| Alt-EI-TS-1 | 1.50 (1.27–1.54) | 7.94e-08 | 18.72 (4–35) | 1.38e-07 | 0.40 |
| Alt-EI-TS-3 | 1.62 (1.28–1.56) | 1.27e-07 | 21.50 (2–33) | 6.32e-08 | 0.38 |
| Alt-EI-TS-5 | 1.58 (1.28–1.58) | 7.78e-08 | 21.16 (3–35) | 1.10e-07 | 0.39 |
| TwoPhases-TS-EI | 1.86 (1.30–1.63) | 6.62e-08 | 24.06 (2–35) | 6.32e-08 | 0.37 |
| *LLM-based Methods* | | | | | |
| LLAMBO | 2.67 (1.14–2.65) | 1.55e-04 | 23.74 (1–38) | 1.49e-06 | 0.43 |
| LLMP | 2.78 (1.59–2.49) | 3.70e-08 | 31.30 (1–38) | 3.52e-08 | 0.33 |
| *Adaptive Portfolio Methods* | | | | | |
| GP-Hedge | 1.45 (1.22–1.52) | 1.82e-07 | 16.96 (1–34) | 1.65e-07 | 0.42 |
| No-PASt-BO | 1.53 (1.20–1.74) | 1.27e-07 | 19.08 (1–37) | 1.65e-07 | 0.37 |
| SETUP-BO | 1.56 (1.21–1.62) | 1.11e-07 | 19.06 (1–37) | 8.91e-08 | 0.42 |
| ESP | 1.62 (1.29–1.67) | 4.30e-08 | 22.98 (1–38) | 5.53e-08 | 0.42 |
| LMABO | **1.21** (1.06–1.25) | – | **5.62** (1–19) | – | 0.37 |
| *P-values of Friedman Tests* | | 1.380e-106 | | 1.380e-106 | |

of about 6000 tokens ($\approx$\$0.01); both this expense and the LLM call latency of roughly 1 second per iteration are negligible relative to the cost of evaluating expensive black-box functions (which often takes minutes or hours per evaluation) and are justified by the resulting performance gains.

Static AFs are inherently unreliable. While strong heuristics like EI or LogEI are among the best in this class with relatively low RPs and ranks, their performance is brittle; on some problems, their rank drops to as low as 35th, highlighting the risk of a fixed strategy. In addition, adaptive portfolio methods, though achieving competitive results, frequently exhibit higher variability and are less robust across the heterogeneous problem suite. These approaches depend on heuristics for weighting past acquisition performance and thus can be slow to adapt when the task-specific land-

Table 2: **Ablation study on the components of LMABO**. We analyze the contribution of LMABO's key components by comparing the full model to multiple ablated versions. LMABO-8B/30B uses open-source LLMs (Qwen3-8B and Qwen3-30B-A3B-Thinking-2507 (Team, 2025)). LMABO-120B uses the open-weight model gpt-oss-120b (OpenAI, 2025). The Mean RP and Mean Rank are calculated using the same global ranking of all baseline and ablation methods as in Table 1.

| Method | Mean RP↓ (Interquartile Range) | P-value (RP) | Mean Rank↓ (Min - Max) | P-value (Rank) | CV of AUC |
|---|---|---|---|---|---|
| *LMABO without:* | | | | | |
| Remaining budget | 1.40 (1.17–1.54) | 4.25e-07 | 15.72 (3–34) | 2.61e-07 | 0.39 |
| GP model characteristics | 1.50 (1.21–1.59) | 4.30e-08 | 20.04 (4–34) | 3.47e-08 | 0.40 |
| Shortest distance information | 1.50 (1.23–1.65) | 1.62e-07 | 19.76 (1–33) | 7.60e-08 | 0.39 |
| Instruction to avoid ineffective AFs | 1.92 (1.45–1.91) | 3.07e-08 | 28.30 (3–37) | 3.02e-08 | 0.43 |
| *LMABO using other LLMs* | | | | | |
| LMABO-8B | 1.48 (1.24–1.62) | 2.48e-07 | 18.94 (1–34) | 1.72e-07 | 0.38 |
| LMABO-30B | 1.29 (1.15–1.35) | 3.21e-04 | 10.70 (2–25) | 6.28e-05 | 0.39 |
| LMABO-120B | 1.22 (1.07–1.24) | 1.00e+00 | 6.64 (1–31) | 3.99e-01 | 0.39 |
| LMABO (GPT-4o mini) | 1.21 (1.11–1.26) | 1.00e+00 | 7.16 (1–21) | 1.84e-01 | 0.37 |
| LMABO | 1.21 (1.06–1.25) | – | 5.62 (1–19) | – | 0.37 |

scape changes or when surrogate uncertainty dynamics differ across problems. Given the same AF portfolio, LMABO mitigates this shortcoming by considering other important factors including the process status, performance history and surrogate model characteristics, which enables faster, more consistent adaptation and yields significantly better average performance. LLM-based methods, while occasionally ranking among the top performers (e.g., LLAMBO is in the top three on 12 out of 50 problems), generally exhibit inconsistent results and often fall behind other approaches. This highlights that simply incorporating an LLM is not sufficient; effective navigation of the exploration-exploitation trade-off is crucial for robust BO. LMABO addresses this by explicitly framing AF selection as a decision-making task in which the LLM can make informed, context-aware decisions to balance the trade-off effectively.

Similar to the aforementioned baselines, simple meta-strategies also struggle to maintain consistent performance across diverse problems. Figures 2a and 2b show that LMABO uses EI, LogEI, and TS more frequently than other AFs and often switches between these three options. However, simple meta-strategies that mimic these behaviors (e.g., random selection between the three or alternating between EI and TS) fail to deliver robust performance. Therefore, these results demonstrate that LMABO's behavior cannot be reduced to a simple heuristic.

## 5.3 ABLATION STUDIES

Our ablation studies confirm that each component of the LMABO framework is crucial for achieving its superior performance, with statistically significant degradation in performance compared to the full model once a component is removed, as shown in Table 2. However, these studies also demonstrate the robustness of our core framework, as the ablated versions still achieve respectable results, often performing on par with or better than many established baselines.

Firstly, LMABO's performance is inherently coupled with the underlying LLM. With a small open-source model in our LMABO-8B experiment, we observed a noticeable performance drop, though it still achieves competitive results. LMABO-30B, using a larger model with improved thinking capabilities, recovers much of this drop, approaching that of the full LMABO with Gemini-2.5 Flash and outperforming all baselines. Both LMABO-120B and LMABO with GPT-4o mini achieve performance comparable to the vanila version, demonstrating that LMABO's effectiveness is not tied to a specific LLM, but rather benefits from the general reasoning capabilities of strong LLMs.

Removing the remaining budget leads to the smallest performance drop, which means that this information is the least critical of the three ablated inputs. GP model characteristics and shortest

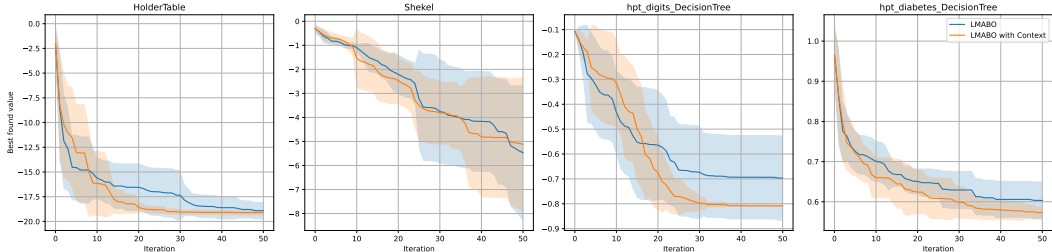

Figure 1: Impact of task-specific context on LMABO performance. Results are averaged over 10 runs with standard deviation shown as shaded regions.

distance information are equivalently impactful on the effectiveness of the LMABO. The large drop in LMABO-AB4 demonstrates that instructing the LLM to avoid AFs that failed to improve the incumbent is essential. Without this guidance, we find that the LLM repeatedly selects ineffective AFs, leading to significantly worse optimization performance.

In addition to these ablation studies, we attempted to inject task-specific context into the initial prompt $P_0$ on 4 benchmark problems, such as characteristics of the synthetic functions or hyperparameter optimization tasks. The hypothesis is that by exploiting prior knowledge about the problem domain, the LLM could make more informed AF selections, potentially enhancing the optimization performance. Details about these contexts are provided in Appendix C.1. The results, shown in Figure 1, suggest that providing context acts as a safeguard against stagnation in local optima, a benefit observed across both synthetic and real-world landscapes. Vanila LMABO may stall at a sub-optimal plateau after the initial progress (e.g. between iterations 10 and 30 on HolderTable); however, being warned of "many local minima", the context-aware variant successfully identified this trap and bypassed it early, converging to the global minimum significantly faster. Based on these findings, we recommend providing a textual description of the objective function whenever such knowledge is available. In hyperparameter optimization scenarios involving well-known algorithms (e.g. Decision Tree or AdaBoost), this semantic warmstarting can significantly reduce the computational budget required to reach competitive performance (which was also reported in Liu et al. (2024) albeit for point initialization).

## 6 ANALYSIS

To understand LMABO's strategy, we performed an in-depth analysis of its AF selection behavior, which reveals a multi-faceted strategy with clear preferences, following distinct temporal patterns, and, most importantly, adapting its behavior in response to the real-time optimization state. Our findings provide strong evidence that *LMABO effectively synthesizes the state information to execute a dynamic, context-aware policy that mirrors established practices in BO*. Note that this section's results are aggregated across all repetitions on all problems from experiments in Section 5.

**Overall Preferences** In Figure 2a, we observe that LMABO exhibits a clear preference for certain AFs (e.g. EI, LogEI, and TS). EI's usage often increases slightly at the beginning as a response to early improvements, while TS's usage decreases gradually near the end as the need for exploration diminishes. PosMean is heavily used near the end for LMABO as a last effort to find improvements. Another surprising observation is the high usage of MES and PES in the first few iterations, but this seems to align with the initial goal of quickly reducing uncertainty about the global optimum. However, the strong performance of LMABO also involves other factors, as discussed later. On other adaptive portfolio methods, there is no clear preference for any AF, and the selection is more uniformly distributed. The only exception is No-PASt-BO with an increasing preference for PosMean as the budget runs out. From this figure, a first insight is that LMABO's success partly stems from a well-calibrated preference for strong AFs, rather than a uniform exploration of all options, as well as a dynamic adjustment of these preferences over time.

On a separate note, the adaptive portfolio baselines can be at a disadvantage without knowing the strengths of EI and LogEI. We conducted an additional experiment where these methods operate on

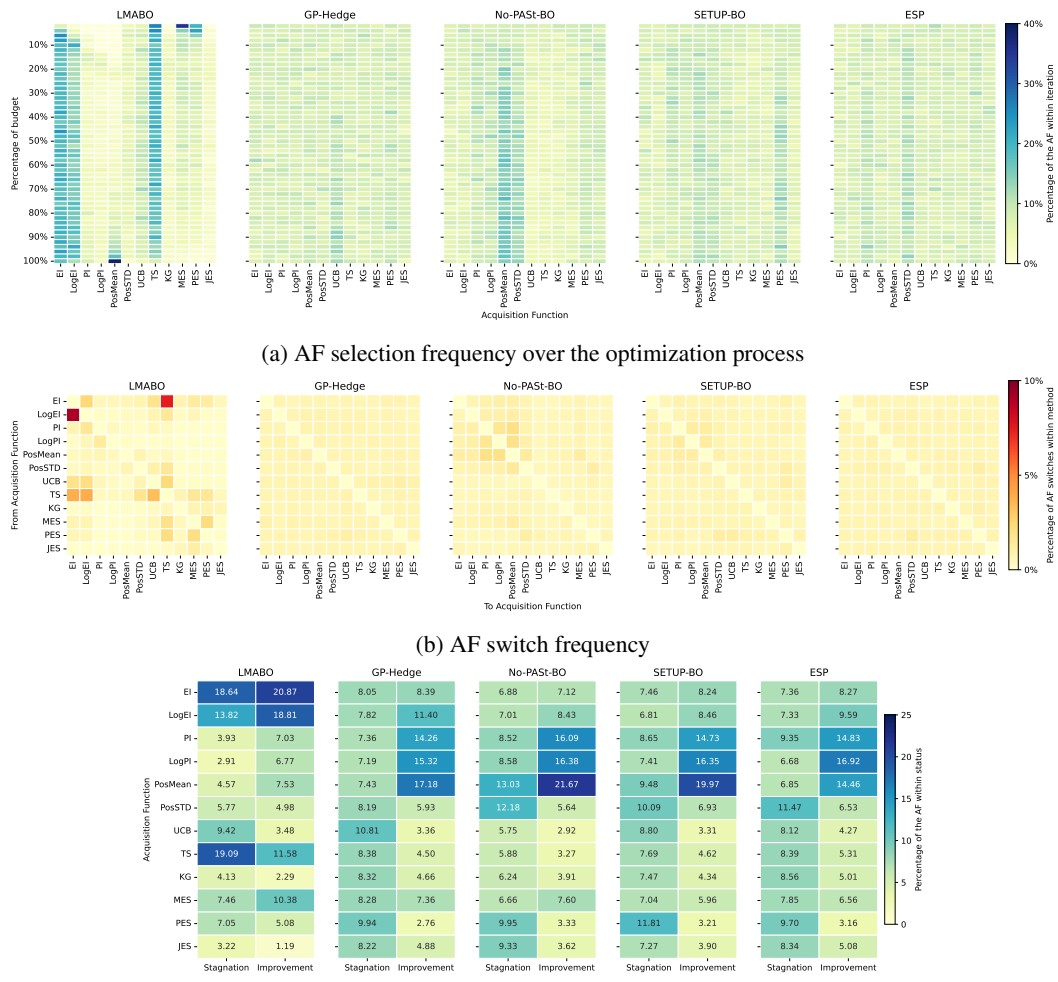

(a) AF selection frequency over the optimization process

(b) AF switch frequency

(c) AF selection frequency by improvement/stagnation status. "Stagnation" is defined as no improvement in the incumbent, and "Improvement" is defined as any change in the incumbent.

Figure 2: LMABO's acquisition function selection behaviors. Note that these behaviors are aggregated across all runs on all problems.

only a curated subset comprising of EI, LogEI, and TS - the three most frequently selected AFs by LMABO. We indeed observe a performance improvement for these methods except for GP-Hedge (detailed results in Appendix E), but the curated versions are still strictly outperformed by LMABO.

**Switching the AF in Response to the Optimization State** In Figure 2b, a notable difference is observed between LMABO and the adaptive portfolio baselines in terms of AF switching. LMABO is the only method to perform a high number of switches between the group of explorative AFs (i.e. the first five AFs) and exploitative AFs (i.e. the remaining seven AFs) throughout the optimization process, as seen in the bottom left and top right of the figures. The dynamic adjustment mentioned earlier is more evident here, as LMABO frequently switches between exploration and exploitation. Figure 2c shows increased usage of exploitative AFs during improvement phases for all methods, which aligns with the goal of refining the search around promising areas. However, combined with the findings from Figure 2b, the adaptive portfolio baselines seem to demonstrate a passive, one-directional learning by using more exploitative functions after success, but their sparse switching patterns reveal a "sticky" policy that is slow to abandon a strategy, even when it is failing. This is likely because of their reliance on past successes to guide future choices, which can lead to overcommitment to a single AF. In contrast, LMABO employs an active, bi-directional strategy, not only learning to exploit on improvement but also decisively switching back to exploratory functions

to escape stagnation. This demonstrates that LMABO's core advantage lies not just in identifying a good heuristic, but in its superior, more agile policy for knowing when that heuristic is no longer effective and a different approach is required.

**Linking AF Selection to Justification**  To verify a consistent link between the LLM's AF choices and its justifications, we analyzed keywords appeared in the justifications across different AFs. In Figure 3, explorative AFs are strongly associated with terms like "exploration" or "stagnation", and exploitative AFs are strongly associated with terms like "exploitation" or "improvement". This confirms that the LLM's selections are consistent with its justification, and that it is possible to use these justfication to understand the decision of the LLM.

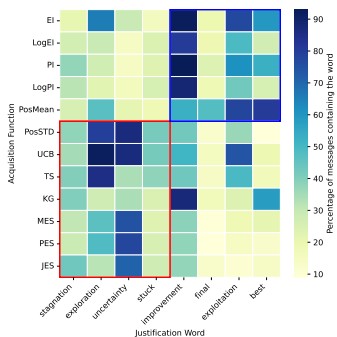

Figure 3: Word frequency in justifications. Red box (bottom left) is the explorative group. Blue box (top right) is the exploitative group.

**Information Sensitivity Analysis**  To assess the sensitivity of LMABO to the information provided in the prompt, we conducted a sensitivity analysis by perturbing the values fed to the LLM in the state representation across early, middle and late stages. Due to space constraints, we summarize the key findings here (detailed results are in Appendix G). During the early stage, the LLM is highly reactive to all information changes, including the process status, performance history, and GP model characteristics. During the middle stage, performance history and process status are most influential. In the late stage, the LLM is less sensitive to perturbations and is mostly changing its decision in response to new incumbent values where it exhibits a strong preference for exploitative AFs. While perturbing the values presents inherent limitations (e.g. the perturbed values may be unrealistic or inconsistent with other state variables), the results still provide valuable insights into the LLM's decision-making process. Overall, we find that LMABO is highly sensitive to tactical variables like performance history with evident signs of stagnation/improvement and process status, while correctly showing robustness to changes in secondary GP parameters that do not alter the overall strategic context. This demonstrates that the LLM is synthesizing the state summary to weigh the relative importance of different inputs, a key feature of its effective, state-aware policy.

## 7 CONCLUSION

We introduced LMABO, a novel framework that successfully utilizes a pre-trained LLM as a zero-shot, online strategist for selecting acquisition functions in Bayesian Optimization. By prompting the LLM with a comprehensive, real-time summary of the optimization state, LMABO dynamically controls the exploration-exploitation trade-off. Our extensive experiments on 50 benchmarks show that this approach significantly outperforms strong static, adaptive, and other LLM-based baselines. Ablation studies and analysis of the LLM's behavior confirm its success stems from a well-rounded state-aware strategy that adapts well to the optimization's progress, demonstrating patterns that align closely with established BO best practices.

This paper focused on standard BO with Gaussian Processes, and future work could adapt LMABO to other surrogate models and optimization settings. For instance, in constrained BO, the LLM could be leveraged to dynamically balance objective improvement against constraint satisfaction. Overall, LMABO opens new avenues for integrating LLMs into adaptive optimization frameworks, leveraging their reasoning capabilities for decision-making in complex tasks.

## REPRODUCIBILITY STATEMENT

Implementation of LMABO and its variants along with the baselines and scripts to reproduce the experiments can be found at `https://github.com/giang-n-ngo/lmabo`.

## LLM USAGE

Large Language Models were used for correcting grammar and improving writing clarity along with updating the related works. We used Gemini Pro and ChatGPT for these purposes.

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

## A  LIST OF ACQUISITION FUNCTIONS

The acquisition function $\alpha(x; \mathcal{D})$ guides the selection of query points in BO by quantifying the utility of evaluating $f$ at $x$. By maximizing $\alpha(x; \mathcal{D})$ over the search space, we identify points that balance exploration (sampling where uncertainty is high) and exploitation (sampling where the surrogate predicts low function values). Details of the AFs used in our experiments are as follows:

- Probability of Improvement (PI) (Kushner, 1964):

$$\alpha_{\text{PI}}(x) = \Phi\left(\frac{\mu(x) - \tau}{\sigma(x)}\right),$$

  where $\Phi$ is the standard normal CDF, $\mu(x)$ and $\sigma(x)$ are the posterior mean and standard deviation, and $\tau$ is a target (e.g., the incumbent solution). PI selects points with high probability of improving upon $\tau$.

- Expected Improvement (EI) (Mockus, 1998):

$$\alpha_{\text{EI}}(x) = (\mu(x) - \tau)\,\Phi(z) + \sigma(x)\,\phi(z), \quad z = \frac{\mu(x) - \tau}{\sigma(x)},$$

  where $\phi$ is the standard normal PDF. EI measures the expected magnitude of improvement.

- Log Probability of Improvement (LogPI) (Balandat et al., 2020) and Log Expected Improvement (LogEI) (Ament et al., 2023): These are numerically stable variants of PI and EI, respectively, that operate in the log domain to handle vanishing values and gradients.

- Upper Confidence Bound (UCB) (Srinivas et al., 2010):

$$\alpha_{\mathrm{UCB}}(x) = \mu(x) + \kappa\, \sigma(x),$$

where $\kappa > 0$ controls the exploration weight. UCB explicitly balances exploitation (mean) and exploration (uncertainty).

- Thompson Sampling (TS) (Chowdhury & Gopalan, 2017): Draws a sample $\tilde{f} \sim p(f \mid \mathcal{D})$ and selects $x$ maximizing $\tilde{f}(x)$. TS provides a randomized exploration strategy consistent with the posterior.

- Posterior Mean (PosMean) and Posterior Standard Deviation (PosStd): Using $\alpha_{\mathrm{mean}}(x) = \mu(x)$ yields pure exploitation, while $\alpha_{\mathrm{std}}(x) = \sigma(x)$ performs pure exploration.

- Knowledge Gradient (KG) (Wu & Frazier, 2016): This look-ahead function quantifies the expected increase in the maximum value of the function that results from collecting a new observation at a candidate point $x$.

- Information-theoretic AFs:

  - Predictive Entropy Search (PES) (Hernández-Lobato et al., 2014): Selects points that maximize the expected reduction in entropy of the distribution over the location of the global optimum.
  - Max-value Entropy Search (MES) (Wang & Jegelka, 2017): Focuses on reducing uncertainty about the maximum function value rather than its location.
  - Joint Entropy Search (JES) (Tu et al., 2022): A recent AF that generalizes PES and MES by considering the joint entropy of both the location and value of the optimum.

We use the implementations of these AFs from the BoTorch library (Balandat et al., 2020) in our experiments. For KG, PES, MES, and JES, we use their available batch implementations with a batch size of 1. We consider PosSTD, UCB, TS, KG, PES, MES, and JES to be in the exploratory category, while the exploitative AFs include PosMean, PI, LogPI, EI, and LogEI.

## B    Experimental Details

### B.1    List of Benchmarks

We evaluate LMABO on a diverse suite of 50 benchmark problems, including synthetic functions and real-world hyperparameter optimization tasks. The synthetic benchmarks are isted in Table 3, with 15 functions from the COCO suite (Hansen et al., 2021) and 15 from BoTorch (Balandat et al., 2020). For hyperparameter optimization tasks, we follow the practice of Liu et al. (2024) and use datasets and ML models available on Bayesmark (Uber, 2020) to form 20 dataset-model pairs. The datasets include Breast, Digits, Wines and Diabetes, and the ML models include Decision Tree, Random Forest, SVM, AdaBoost, and MLPSGD. The dimensionality of these search spaces ranges from 2 to 8. We use the empirical optimum found by all methods (as the true optima are unknown for many problems) to compute the simple regret at each iteration. See Appendix D.1 of Liu et al. (2024) for full details about the hyperparameter optimization tasks.

### B.2    Implementation Details

We follow the standard practice to optimize the GP hyperparameters by maximizing the log marginal likelihood at each iteration. We use the default implementation of GP regression in BoTorch (Balandat et al., 2020) which has hyperparameter priors from (Hvarfner et al., 2024) to enhance performance on high-dimensional tasks. All acquisition functions are optimized using multi-start LBFGS-B, except for TS and PES. For both TS and PES, we use discrete optimization over a finite set of randomly sampled candidates due to the high computational cost of evaluating the acquisition functions. This practice of AF optimization applies to running experiments with all static acquisition functions, all adaptive portfolio baselines, and LMABO.

Table 3: COCO and BoTorch synthetic benchmark functions used in our experiments.

| COCO benchmarks (15) | | BoTorch synthetic benchmarks (15) | |
|---|---|---|---|
| Name | Dimensionality | Name | Dimensionality |
| BucheRastrigin | 5 | Ackley | 50 |
| LinearSlope | 5 | Beale | 2 |
| AttractiveSector | 5 | Bukin | 2 |
| StepEllipsoid | 5 | Cosine8 | 8 |
| Discus | 5 | DixonPrice | 15 |
| BentCigar | 5 | DropWave | 2 |
| SharpRidge | 5 | EggHolder | 2 |
| DifferentPowers | 5 | Griewank | 9 |
| Weierstrass | 5 | Hartmann | 6 |
| SchaffersIllCond | 5 | HolderTable | 2 |
| CompositeGriewankRosenbrock | 10 | Levy | 13 |
| Gallagher21 | 5 | Michalewicz | 10 |
| Gallagher101 | 5 | StyblinskiTang | 21 |
| Katsuura | 5 | Shekel | 4 |
| LunacekBiRastrigin | 5 | SixHumpCamel | 2 |

For all experiments, the LLMs were queried with a temperature of 0.0. Again, for invalid or failed LLM responses, we fall back to UCB. The fallback rate is extremely small at only about 0.11% of all queries (i.e. out of 10,000 queries, only 11 of them do not follow the "Acquistion abbreviation: justfication" format). This indicates that LLM failures are rare and have minimal impact on overall optimization performance. All adaptive portfolio baselines, such as GP-Hedge, were implemented using their standard configurations as described in their respective publications, except that the portfolio of acquisition functions was expanded to include all 12 AFs listed in Appendix A.

## B.3 STATISTICAL TESTS

Results in Tables 1, 2 and 6 follow a standardized statistical analysis to ensure the significance of the results. This statistical analysis is conducted separately for both mean RPs and ranks (across 10 repetitions) of 38 methods on 50 benchmarks. Firstly, to assess the significance of performance differences between the methods, we first apply the Friedman test (a non-parametric test for data that is not normally distributed) on the matrix of mean RPs (or ranks). The null hypothesis of the Friedman test is that all methods perform equally, while the alternative hypothesis is that at least one method performs differently. If the Friedman test indicates significant differences, we follow up with post-hoc pairwise comparisons using the Wilcoxon signed-rank test with Holm-Bonferroni correction to control for multiple comparisons, specifically comparing each baseline method against LMABO. The null hypothesis for each pairwise comparison is that there is no difference in performance between LMABO and a baseline, while the alternative hypothesis is that the performances of LMABO and the baseline differ. We set a significance level of 0.05 for all statistical tests.

## C PROMPTS

Figure 1 shows the complete, unabridged initial prompt ($P_0$) used to instruct the LLM in our experiments. This prompt was developed through a very brief iterative process of 6-7 trials. The refinements were not aimed at tuning for performance on a specific benchmark, but rather to ensure accurate formatting of the LLM's responses and to encourage a full consideration of all information in the optimization state representation. Examples of follow up prompts during the optimization process are shown in Table 8. For KG, PES, MES, and JES, we denote them by qKG, qPES, qMES, and qJES, respectively, in the initial prompt to align with the naming conventions in BoTorch (Balandat et al., 2020). In constructing the initial prompt, we observed the following phenomena during preliminary experiments that informed the final design:

Table 4: Additional contexts provided to the LLM to verify the benefit of prior knowledge on LMABO.

| Problem | Context |
|---|---|
| HolderTable | This is to optimize a black-box synthetic function defined on a 2-dimensional bounded domain. The landscape is highly complex and non-convex, featuring a wavy pattern with many local minima. My prior analysis strongly suggests that the global minimum is not unique. |
| Shekel | This is to optimize a black-box synthetic function defined on a 4-dimensional bounded domain. The landscape should be mostly flat, but it is punctuated by a small number of sharp, narrow, and deep hollows at some locations. The main challenge is not finding the general region of these minima, but precisely pinpointing the 'needle in a haystack' global minimum at the very bottom of one of these steep basins. |
| hpt_digits_DecisionTree | This is to optimize a 6-dimensional space for a Decision Tree classifier on the Digits dataset, where the objective is to maximize validation accuracy (by minimizing the negated values). The landscape is relatively smooth but features several ridges and valleys due to the complex interactions between parameters like max_depth, max_features, and min_samples_split. The main challenge lies in balancing these parameters to avoid overfitting while still capturing the underlying patterns in the data. My analysis indicates that certain combinations of these parameters can lead to similar accuracy levels, suggesting multiple optimal regions in the parameter space. |
| hpt_diabetes_DecisionTree | This is to optimize a 6-dimensional space for a Decision Tree regressor on the Diabetes dataset, where the objective is to minimize mean squared error. The landscape is relatively smooth but features several ridges and valleys due to the complex interactions between parameters like max_depth, max_features, and min_samples_split. The main challenge lies in balancing these parameters to avoid overfitting while still capturing the underlying patterns in the data. My analysis indicates that certain combinations of these parameters can lead to similar error levels, suggesting multiple optimal regions in the parameter space. |

- A full history of past function values and AF choices was not necessary for each input prompt, as the LLM seems capable of inferring relevant optimization history from previous input prompts.

- Specific examples of AF choices (e.g. "UCB: brief justification") in the prompt could bias the LLM towards a particular choice, so we replaced them with placeholders. Before this change, we found that Qwen3-8B often mimicked the exemplar choices if specific AFs were mentioned as examples, and only stopped doing so when the examples were replaced with placeholders.

- The LLM sometimes ignored certain details, such as the number of remaining iterations, so we added an assurance in the prompt to consider all provided context.

### C.1    ADDITIONAL CONTEXTS

Table 4 contains the task-specific contexts provided to the LLM to test how such prior knowledge about the problem would affect LMABO's performance. These contexts were added to the initial prompt $P_0$ between lines 3 and 5 in Figure 1.

## D    RUNTIME COMPARISON

Table 5 shows the average runtime, in minutes, for 50 iterations of the methods across all tested benchmarks. LMABO only incurs a moderate overhead compared to static AFs due to the LLM

query at each iteration, but it is significantly faster than adaptive portfolio methods that require optimizing multiple AFs at each iteration. Since the curated versions of adaptive portfolio methods only optimize 3 AFs instead of 12, they are much faster than their full versions while achieving better performance (as shown in Table 6), suggesting that a smaller, well-chosen portfolio can be beneficial for these methods. Among LLM-based methods, LLAMBO is comparable in speed to LMABO, while LLMP is much slower due to inferring with an open-source model locally. This higher runtime is also observed on LMABO-8B/30B/120B, the ablated versions of LMABO that use an open-source model.

Table 5: Average runtime for 50 iterations of all methods across all benchmarks (in minutes).

| Method | Runtime | Method | Runtime |
|--------|---------|--------|---------|
| PosSTD | 2.20 | GP-Hedge-Curated | 12.01 |
| PosMean | 2.06 | No-PASt-BO | 113.67 |
| PI | 2.62 | No-PASt-BO-Curated | 14.94 |
| LogPI | 2.09 | SETUP-BO | 104.42 |
| EI | 2.14 | SETUP-BO-Curated | 7.24 |
| LogEI | 2.15 | ESP | 50.61 |
| UCB | 2.07 | ESP-Curated | 6.01 |
| TS | 4.62 | LMABO (with Gemini 2.5 Flash/GPT-4o mini) | 7.42/7.36 |
| KG | 12.88 | LMABO-AB1 | 6.69 |
| PES | 8.93 | LMABO-AB2 | 8.17 |
| MES | 3.73 | LMABO-AB3 | 7.79 |
| JES | 5.45 | LMABO-AB4 | 6.46 |
| LLAMBO | 9.21 | LMABO-8B | 19.12 |
| LLMP | 29.35 | LMABO-30B | 14.87 |
| GP-Hedge | 109.18 | LMABO-120B | 15.61 |

## E    CURATED SET FOR ADAPTIVE PORTFOLIO BASELINES

In this experiment, the adaptive portfolio baselines (i.e. GP-Hedge, No-PASt-BO, SETUP-BO, and ESP) are restricted to a curated subset of acquisition functions: EI, LogEI, and TS. The curated methods also participated in the calculation of relative performance, rank, and statistical tests mentioned in Section 5. No-PASt-BO, SETUP-BO, and ESP show a performance improvement when using the curated set, while GP-Hedge shows a degradation.

## F    LLM RESPONSE EXAMPLES

### F.1    RESPONSES TO THE INITIAL PROMPT

Table 7 shows the responses of Gemini-2.5 Flash to the initial prompt across some different optimization problems. The responses confirm the LLM's understanding of the task and readiness to proceed with the optimization process.

### F.2    RESPONSES DURING OPTIMIZATION

Table 8 shows example responses from Gemini-2.5 Flash at different stages of an optimization run of the Weierstrass function. As seen on the table, the LLM adapts its acquisition function choices based on all provided context, including the number of remaining iterations, the current best objective value, the model's lengthscales, and the shortest distance between points. Contrary to the popular choice of TS in early iterations in other cases, the first response opts for EI when it observes a wide range of function values. The second and fourth responses both select LogEI after a new best value is found, showing the LLM's ability to recognize when an improvement-focused acquisition is appropriate. Although both EI and LogEI are suitable in these contexts (as well as other exploitative AFs like PI), the LLM's choice of LogEI is influenced by the wide range of function values and the modest gain compared to the range (which were from -6.576 to -9.980 for the second response

Table 6: Comparing adaptive portfolio methods between using a large portfolio (of 12 AFs) and a curated portfolio (of 3 AFs). The curated methods are denoted with a "-Curated" suffix.

| Method | Mean RP↓ (Interquartile Range) | P-value (RP) | Mean Rank↓ (Min - Max) | P-value (Rank) | CV of (AUC) |
|---|---|---|---|---|---|
| GP-Hedge | 1.422 (1.206–1.517) | 1.239e-07 | 12.42 (1–25) | 1.112e-07 | 0.418 |
| GP-Hedge-Curated | 1.487 (1.233–1.670) | 3.126e-08 | 15.06 (2–28) | 5.444e-08 | 0.372 |
| No-PASt-BO | 1.505 (1.186–1.722) | 9.572e-08 | 14.06 (1–28) | 1.112e-07 | 0.370 |
| No-PASt-BO-Curated | 1.469 (1.218–1.579) | 7.656e-08 | 14.38 (1–24) | 7.235e-08 | 0.409 |
| SETUP-BO | 1.536 (1.155–1.688) | 8.609e-08 | 13.88 (1–28) | 6.129e-08 | 0.420 |
| SETUP-BO-Curated | 1.503 (1.243–1.556) | 3.126e-08 | 14.76 (1–26) | 4.781e-08 | 0.374 |
| ESP | 1.601 (1.273–1.674) | 3.126e-08 | 16.82 (1–29) | 3.742e-08 | 0.419 |
| ESP-Curated | 1.506 (1.240–1.537) | 9.572e-08 | 15.06 (2–26) | 1.112e-07 | 0.422 |

Table 7: Responses to the initial prompt $P_0$.

| Problem | Response |
|---|---|
| Cosine8 | Yes, I understand the context. I am ready to receive the first summary of the Bayesian Optimization process. |
| Weierstrass | Okay, I understand. I am ready to receive the summary of the Bayesian Optimization process and recommend the next acquisition function. I will strictly follow the specified output format. |
| hpt_wine_MLPSGD | Yes, I understand the task and the available acquisition functions. I am ready to receive the first summary of the Bayesian Optimization process. |

and from -10.148 to -11.302 for the fourth response). This is helpful when the values of EI may become very small, as LogEI is more numerically stable. The third response chooses TS to escape a stagnation phase in response to failed improvements and over-exploration signs (demonstrated by the smaller shortest distance). Finally, with only one iteration left, the LLM selects EI to maximize the chance of a final improvement.

## G   INFORMATION SENSITIVITY ANALYSIS

Tables 9, 10, 11, and 12 show the results of our information sensitivity analysis at early, middle, and late stages of optimization, respectively. For these experiments, we perturb each element of the state representation $S_t$ individually while keeping all other elements fixed to their original values at iteration $T$ in an optimization run. Input prompts from previous iterations were not modified, so the LLM's memory of the optimization history remains intact.

While perturbing the values may present some noises in the results, we still observe some clear trends across all four tables. Firstly, information about the process status (e.g. number of points evaluated, remaining budget) and the performance history (e.g. incumbent objective value, function value range, shortest distance) have a significant impact on the LLM's acquisition function choices. Reducing the remaining budget or having a new incumbent objective value tends to shift the LLM's preference towards exploitative AFs. However, in Tables 9 and 10, we observe that a very small improvement in the incumbent objective value (e.g. from 1.24 to 1.244 in Table 9 and from -9.98 to -9.982 in Table 10) can lead to a switch back to an explorative AF. This suggests that the LLM is sensitive to the magnitude of improvement relative to the overall function value range. Secondly, information about the model state (e.g. lengthscales, outputscale) appears to have a more subtle influence on the LLM's choices. While perturbing these elements does lead to some changes in the selected AFs, the changes are less consistent and pronounced compared to the other state elements. This indicates that the LLM may prioritize information about the optimization progress and performance over the surrogate model's internal parameters when making decisions.

```
1   You are an expert in Bayesian Optimization, specifically tasked with
    recommending the most suitable acquisition function for the next
    iteration to minimize an objective function.
2
3   For context, we use a Gaussian Process as the surrogate model with a
    Matern 5/2 kernel with ARD.
4
5   I will provide you with a summary of the Bayesian Optimization process at
     each step. This summary will include the following information:
6   - **N:** The total number of points evaluated so far.
7   - **Remaining iterations:** The number of iterations left in the
    optimization process.
8   - **D:** The dimensionality of the search space (number of input
    parameters).
9   - **f_range:** The range of the objective function values observed so far
    .
10  - **f_min:** The current best (lowest) observed objective value.
11  - **Shortest distance:** The shortest distance from the last point to any
     other point, indicating whether it is exploiting too much.
12  - **Model lengthscales:** These are crucial hyperparameters of the
    Gaussian Process model's kernel.
13  They describe how the model perceives the smoothness and relevance of
    each input dimension to the objective function.
14  You will receive their range (min/max), mean, and standard deviation.
15  - **Model outputscale:** It defines the overall magnitude or amplitude of
     the function's variation.
16
17  Available acquisition functions you can choose from, with brief
    descriptions of their primary uses:
18  1.  PI (Probability of Improvement)
19  2.  LogPI (Log Probability of Improvement)
20  3.  EI (Expected Improvement)
21  4.  LogEI (Log Expected Improvement)
22  5.  UCB (Upper Confidence Bound)
23  6.  PosMean (Posterior Mean):
24  7.  PosSTD (Posterior Standard Deviation)
25  8.  TS (Thompson Sampling)
26  9.  qKG (Knowledge Gradient)
27  10. qPES (Predictive Entropy Search)
28  11. qMES (Max-value Entropy Search)
29  12. qJES (Joint Entropy Search)
30
31  At each step:
32  - **Review the provided summary of the optimization process and consider
    the current state of the optimization.**
33  - **Select the acquisition function that you believe will be best for the
     optimization process.**
34  - **Avoid reusing acquisition functions that failed to improve the
    objective function in previous iterations.**
35
36  When responding, select the acquisition function you deem most
    appropriate.
37  Your justification should briefly explain why that function is suitable
    given the provided optimization summary, referencing relevant aspects
    like exploration/exploitation balance, remaining iterations, or model
    characteristics.
38  The response must strictly follow the format "Acquisition abbreviation:
    justification", similar to these examples:
39  - 'AF_ABBREVIATION: Your justification for choosing this specific
    function.'
40  - 'XXX: A brief reason explaining why XXX is the optimal choice now.'
41  Firstly, just give a brief confirmation that you understand the task and
    the available acquisition functions.
```

Listing 1: The unabridged initial prompt ($P_0$) provided to the LLM.

| Input Prompt | Response |
| --- | --- |
| Current optimization state:
- N: 11
- Remaining iterations: 50
- D: 5
- f_range: Range [-5.982, 83.765], Mean 26.466 (Std Dev 35.075)
- f_min: -5.982
- Shortest distance: 0.455
- Lengthscales: Range [0.261, 59.707], Mean 27.581 (Std Dev 24.557)
- Outputscale: 0.898 | EI: Given the early stage (N=11, 50 iterations remaining) and the wide range of observed function values and varying lengthscales, there is still significant uncertainty and potential for discovering better regions. EI provides a robust balance between exploration (considering areas of high uncertainty) and exploitation (seeking expected improvement over the current best, f_min), making it an excellent choice for building a better global understanding of the objective function while still prioritizing improvement. |
| Current optimization state:
- N: 21
- Remaining iterations: 40
- D: 5
- f_range: Range [-9.980, 106.046], Mean 23.896 (Std Dev 34.093)
- f_min: -9.980
- Shortest distance: 0.427
- Lengthscales: Range [0.513, 399.250], Mean 154.676 (Std Dev 188.785)
- Outputscale: 1.212 | LogEI: Thompson Sampling successfully discovered a new f_min. With this new best observation and 40 iterations remaining, LogEI is a strong choice. It is effective for finding further significant improvements, especially when dealing with a potentially wide range of objective values (as indicated by f_range) or when the current best is challenging to beat. LogEI provides a more robust measure of improvement compared to standard EI, by focusing on the log of the improvement, which can help in navigating the complex landscape suggested by the varying lengthscales and ensure continued progress. |
| Current optimization state:
- N: 40
- Remaining iterations: 21
- D: 5
- f_range: Range [-10.148, 132.117], Mean 15.999 (Std Dev 33.416)
- f_min: -10.148
- Shortest distance: 0.204
- Lengthscales: Range [0.449, 3.865], Mean 1.786 (Std Dev 1.159)
- Outputscale: 1.618 | TS: Both JES and PES failed to improve f_min. The lengthscales have drastically shortened and become concentrated, indicating the model believes the function is very localized, yet this hasn't yielded improvements. The shortest distance is also very small (0.20), suggesting potential over-exploitation. With 21 iterations remaining, Thompson Sampling is the best choice to break this persistent stagnation and localized search. By sampling a new function from the posterior, TS can explore regions that the current model might be overlooking or prematurely discarding, offering a fresh, diverse exploration strategy to find new global optima. |
| Current optimization state:
- N: 50
- Remaining iterations: 11
- D: 5
- f_range: Range [-11.302, 132.117], Mean 12.749 (Std Dev 30.799)
- f_min: -11.302
- Shortest distance: 0.346
- Lengthscales: Range [0.494, 2.198], Mean 1.532 (Std Dev 0.611)
- Outputscale: 2.069 | LogEI: EI successfully improved f_min! With a new best value found and only 11 iterations remaining, it's crucial to build on this success with an improvement-focused acquisition function. LogEI is an excellent choice as it continues to seek expected improvement, but its logarithmic scaling makes it particularly robust to the magnitude of objective function values. This allows it to efficiently pinpoint further significant improvements or refine the search around the new f_min, ensuring we maximize the chances of finding an even better optimum in these final, critical iterations. |
| Current optimization state:
- N: 60
- Remaining iterations: 1
- D: 5
- f_range: Range [-12.135, 132.117], Mean 11.070 (Std Dev 29.387)
- f_min: -12.135
- Shortest distance: 0.503
- Lengthscales: Range [0.548, 2.468], Mean 1.615 (Std Dev 0.653)
- Outputscale: 2.821 | EI: With only 1 iteration remaining, the primary goal is to make the absolute best attempt to improve the current f_min. EI (Expected Improvement) is the most robust and direct improvement-focused acquisition function. It directly optimizes for the expected gain over the current best observed value, offering the most pragmatic strategy to find a final, better optimum, especially since it demonstrated success in finding the most recent f_min just a few iterations ago. |

Table 8: Example responses from the LLM during an optimization run of the Weierstrass function. The left column includes the input prompts fed to the LLM. Each response includes the LLM's selected acquisition function along with its justification.

Table 9: Information Sensitivity Analysis for Early Stage ($T = 5$). We perturb the values of each element in the state representation $S_t$ and observe the resulting changes in the output AF chosen by the LLM. The highlighted values indicate the original state element values at $T = 5$ in a Griewank optimization run. We show explorative AFs in blue and exploitative AFs in magenta for better tracking of AF changes between the two groups. The original input prompt and response are shown at the bottom of the table.

| State element | | Output AF given element value | | | | | |
|---|---|---|---|---|---|---|---|
| # points evaluated | Value | 5 | **9** | 20 | 40 | 50 | 500 |
| | AF | PES | LogEI | LogEI | PosSTD | PosSTD | PES |
| Remaining budget | Value | 1 | 10 | 40 | **46** | 50 | 100 |
| | AF | PosMean | LogEI | PES | LogEI | PES | PES |
| Incumbent objective value | Value | -100 | -10 | 0 | 1 | 1.24 | **1.244** |
| | AF | LogEI | EI | EI | EI | KG | LogEI |
| Maximum function value | Value | 2 | 100 | 190 | **194.081** | 200 | 1000 |
| | AF | LogEI | LogEI | UCB | LogEI | LogEI | PES |
| Mean of function value | Value | 2 | 20 | 50 | **56.326** | 60 | 100 |
| | AF | PosSTD | UCB | LogEI | LogEI | LogEI | LogEI |
| Std Dev of function value | Value | 1 | 10 | 60 | **63.145** | 70 | 100 |
| | AF | JES | PES | PES | LogEI | PES | PES |
| Shortest distance | Value | 0.01 | 0.05 | **0.06** | 0.07 | 0.1 | 0.5 |
| | AF | PES | LogEI | LogEI | LogEI | LogEI | MES |
| Minimum of lengthscales | Value | 0.01 | 0.1 | 0.2 | **0.231** | 0.3 | 0.45 |
| | AF | LogEI | PES | LogEI | LogEI | LogEI | PES |
| Maximum of lengthscales | Value | 0.25 | 0.4 | **0.452** | 0.5 | 1.0 | 10.0 |
| | AF | LogEI | EI | LogEI | PES | LogEI | EI |
| Mean of lengthscales | Value | 0.24 | 0.3 | **0.342** | 0.4 | 0.45 | |
| | AF | MES | LogEI | LogEI | LogEI | LogEI | |
| Std Dev of lengthscales | Value | 0.01 | 0.1 | **0.110** | 0.2 | 1.0 | 10.0 |
| | AF | PES | PES | LogEI | LogEI | LogEI | JES |
| Outputscale | Value | 0.01 | 0.1 | 0.8 | **0.865** | 0.9 | 10.0 |
| | AF | LogEI | LogEI | LogEI | LogEI | EI | KG |

| **Original input prompt** | **Original response** |
|---|---|
| Current optimization state: 
 - N: 9 
 - Remaining iterations: 46 
 - D: 2 
 - f_range: Range [1.244, 194.081], Mean 56.326 (Std Dev 63.145) 
 - f_min: 1.244 
 - Shortest distance: 0.060 
 - Lengthscales: Range [0.231, 0.452], Mean 0.342 (Std Dev 0.110) 
 - Outputscale: 0.865 | LogEI: 'f_min' has stalled for two iterations and the shortest distance is again very low (0.060), indicating potential over-exploitation or getting stuck in a flat region. LogEI is suitable here because it is less sensitive to the absolute magnitude of improvement and focuses on relative improvements, which helps to escape local minima or flat regions where small absolute improvements are still valuable, while also maintaining a strong exploration component given the many remaining iterations. We also need to avoid reusing EI which was used 2 iterations ago and did not improve f_min. |

Table 10: Information Sensitivity Analysis for Middle Stage ($T = 25$). This table follows the same format as Table 9 but for $T = 25$ in a CompositeGriewankRosenbrock optimization run.

| State element | | Output AF given element value | | | | | |
|---|---|---|---|---|---|---|---|
| # points evaluated | Value | 5 | 20 | 40 | 45 | 50 | 100 |
| | AF | PI | PI | KG | TS | JES | LogPI |
| Remaining budget | Value | 1 | 5 | 20 | 26 | 30 | 100 |
| | AF | PI | LogEI | KG | TS | TS | LogPI |
| Incumbent objective value | Value | -1000 | -200 | -100 | -92 | -91.2 | -91.176 |
| | AF | PosMean | PI | LogPI | JES | JES | TS |
| Maximum function value | Value | 1 | 20 | 200 | 208.249 | 210 | 1000 |
| | AF | JES | TS | KG | TS | JES | UCB |
| Mean of function value | Value | -90 | -50 | -40.161 | -40 | 0 | 200 |
| | AF | PES | PES | TS | PES | TS | TS |
| Std Dev of function value | Value | 1 | 10 | 50 | 55.658 | 60 | 100 |
| | AF | JES | TS | TS | TS | JES | KG |
| Shortest distance | Value | 0.01 | 0.1 | 0.7 | 0.714 | 0.8 | 1.0 |
| | AF | UCB | TS | JES | TS | UCB | TS |
| Minimum of lengthscales | Value | 0.01 | 0.1 | 0.9 | 0.989 | 1.0 | 10.0 |
| | AF | JES | JES | JES | TS | KG | KG |
| Maximum of lengthscales | Value | 30.0 | 100.0 | 200.0 | 208.249 | 210 | 1000.0 |
| | AF | JES | PES | JES | TS | JES | TS |
| Mean of lengthscales | Value | 1.0 | 10.0 | 20.0 | 29.547 | 30.0 | 200.0 |
| | AF | JES | JES | JES | TS | PES | JES |
| Std Dev of lengthscales | Value | 0.1 | 10.0 | 60.0 | 62.256 | 70.0 | 200.0 |
| | AF | KG | TS | KG | TS | KG | JES |
| Outputscale | Value | 0.1 | 3.0 | 3.731 | 4.0 | 10.0 | 100.0 |
| | AF | UCB | PES | TS | KG | JES | PES |

**Original input prompt**

Current optimization state:
- N: 45
- Remaining iterations: 26
- D: 10
- f_range: Range [-91.176, 208.249], Mean -40.161 (Std Dev 55.658)
- f_min: -91.176
- Shortest distance: 0.714
- Lengthscales: Range [0.989, 204.500], Mean 29.547 (Std Dev 62.256)
- Outputscale: 3.731

**Original response**

TS: 'f_min' remains stagnant for four iterations, and 'qMES' did not yield an improvement, despite the refined lengthscales. The lengthscales have slightly widened again, indicating some lingering uncertainty. With a significant number of remaining iterations and persistent stagnation, 'TS' (Thompson Sampling) is a strong choice. It provides a robust, probabilistic, and fundamentally different exploration strategy by sampling from the GP posterior. This approach is excellent for escaping local optima and discovering genuinely new, high-potential regions, especially when other more deterministic or information-theoretic methods have struggled to find a better 'f_min', and the model still has some uncertainty in its overall landscape understanding.

Table 11: Information Sensitivity Analysis for Late Stage ($T = 45$) - Part 1. This table follows the same format as Table 9 but for $T = 45$ in a hpt_breast_MLPSGD optimization run.

| State element | | Output AF given element value | | | | | |
|---|---|---|---|---|---|---|---|
| # points evaluated | Value | 5 | 20 | 50 | 54 | 60 | 100 |
| | AF | LogPI | JES | JES | JES | JES | JES |
| Remaining budget | Value | 1 | 5 | 6 | 10 | 50 | 100 |
| | AF | EI | KG | JES | JES | JES | JES |
| Incumbent objective value | Value | -100 | -10 | -1 | -0.92 | -0.917 | -0.916 |
| | AF | EI | EI | EI | EI | LogEI | JES |
| Maximum function value | Value | -0.7 | -0.4 | -0.360 | -0.3 | -0.1 | 0.0 |
| | AF | JES | KG | JES | JES | JES | JES |
| Mean function value | Value | -0.9 | -0.8 | -0.738 | -0.7 | -0.4 | |
| | AF | JES | JES | JES | KG | JES | |
| Std Dev function value | Value | 0.01 | 0.1 | 0.152 | 0.2 | 0.5 | 1.0 |
| | AF | JES | JES | JES | JES | JES | JES |
| Shortest distance | Value | 0.01 | 0.1 | 0.4 | 0.459 | 0.5 | 1.0 |
| | AF | JES | JES | JES | JES | KG | JES |

Table 12: Information Sensitivity Analysis for Late Stage ($T = 45$) - Part 2. Continuation of Table 11.

| State element | | Output AF given element value | | | | | |
|---|---|---|---|---|---|---|---|
| Minimum of lengthscales | Value | 0.001 | 0.004 | 0.01 | 0.1 | 1.0 | 10.0 |
| | AF | JES | JES | KG | JES | JES | JES |
| Maximum of lengthscales | Value | 1600 | 2000 | 4000 | 4234.465 | 5000 | 10000 |
| | AF | JES | KG | JES | JES | JES | JES |
| Mean of lengthscales | Value | 1.0 | 100.0 | 1500.0 | 1514.580 | 1600 | 4000 |
| | AF | JES | JES | JES | JES | JES | KG |
| Std Dev of lengthscales | Value | 10.0 | 100.0 | 1600.0 | 1683.459 | 1700.0 | 4000.0 |
| | AF | JES | JES | JES | JES | JES | JES |
| Outputscale | Value | 0.01 | 0.1 | 0.8 | 0.818 | 0.9 | 10.0 |
| | AF | JES | JES | JES | JES | JES | JES |

| Original input prompt | Original response |
|---|---|
| Current optimization state: 
 - N: 54 
 - Remaining iterations: 6 
 - D: 6 
 - f_range: Range [-0.916, -0.360], Mean -0.738 (Std Dev 0.152) 
 - f_min: -0.916 
 - Shortest distance: 0.459 
 - Lengthscales: Range [0.004, 4234.465], Mean 1514.580 (Std Dev 1683.459) 
 - Outputscale: 0.818 | qJES: 'f_min' remains unchanged, meaning qMES did not immediately lead to an improvement. The lengthscales are still extremely large and variable (Range [0.004, 4234.465], Mean 1514.580, Std Dev 1683.459), indicating persistent, profound uncertainty in the GP model's understanding of the objective function. With only 6 iterations remaining (N=54), and 'f_min' still stagnant despite various information-theoretic methods, a comprehensive information-gathering strategy is needed to make the best final decision. qJES (Joint Entropy Search) is an advanced acquisition function that directly aims to reduce the entropy of the *joint* posterior distribution of the optimal value *and* its location. In these very late stages with high model uncertainty, a holistic understanding of both the value and location of the optimum is crucial for making the final, most informed decision. qJES provides a more complete information gain than qPES or qMES alone, making it ideal for the limited remaining budget to resolve uncertainty about the true optimum. (qJES was used at N=32, and its strength in holistic uncertainty reduction makes it appropriate for this critical, late-stage, high-uncertainty scenario). |

