# OpenReview forum: "Adaptive Acquisition Selection for Bayesian Optimization with Large Language Models"
_ICLR.cc/2026/Conference — ICLR 2026 Poster_

### Official Review · Reviewer_Sjvn · 2025-10-28

**Soundness:** 3
**Presentation:** 4
**Contribution:** 2
**Rating:** 6
**Confidence:** 3

**Summary:**

This paper introduces Language Model-assisted Adaptive Bayesian Optimization (LMABO), a policy for selecting, at each iteration of a BO algorithm, the most promising acquisition function to optimize for finding the next observation. LMABO involves a portfolio of acquisition functions and a Large Language Model (LLM) that is tasked to pick the most promising acquisition function given a prompt that summarizes the current optimization state.

LMABO is evaluated against popular acquisitions functions, state-of-the-art strategies for acquisition function selection and heuristics on 50 benchmarks. The numerical results show a statistically significant improvement over the state-of-the-art.

**Strengths:**

* A new strategy for selecting an acquisition function would likely spark interest in the BO community.

* The paper is easy to read and well-written.

* Extensive numerical evaluation with relevant statistical tests.

**Weaknesses:**

* The main weakness is probably the lack of understanding of how the LLM takes its decision. Although LMABO requires a justification from the LLM, there is no guarantee that this justification is actually valid. In that sense, I think an additional control solution, consisting of picking the acquisition function uniformly at random in the acquisition function portfolio, would be very interesting to consider.

* Another worry that I have is that LMABO builds on implicit knowledge of the LLM about acquisition functions in the portfolio. Therefore, integrating a brand new acquisition function in LMABO would probably prove to be challenging, as the LLM would not have any prior knowledge about it.

* Minor presentation issues: a few typos (e.g., "miultiple" on line 125) and the captions of Figure 1(c) and 1(d) are hard to read.

**Questions:**

Here are a few questions to spark the discussion with the authors.

* Do you know how a simple control strategy that picks an acquisition function at random in the portfolio would behave? What about a strategy that randomly picks one of the most popular acquisition functions (e.g., a reduced portfolio with EI, TS, UCB, PosMean)?

* How would you plan to add a new, unknown acquisition function to the portfolio?

* For each real-world experiment, have you tried to exploit some prior knowledge of the LLM on this particular task? For example, for an hyperparameter optimization task involving a gradient step $\epsilon$ and a batch size $b$, have you adapted the initial prompt $P_0$ to describe this particular problem and to indicate to the LLM that it was working on the optimization of a gradient step and a batch size?

---

> ### Author Response · Authors · 2025-11-23
>
> We thank the reviewer for the constructive comments. Below are our responses to the main points raised.
> **Regarding how the LLM takes its decision**
> We share the reviewer's interest in validating that the LLM's decisions are grounded in the optimization state rather than being arbitrary.
> To address this, we have incorporated the suggested randomized control experiment, while our analysis in Section 6 already provides insights into the LLM's decision-making process.
> 1. **Comparison against Random Baselines**
>     We evaluated multiple random baselines (e.g., random with full portfolio, random with a portfolio of popular AFs, and random with a portfolio of top 3 AFs selected by LMABO), and results for these experiments are shown in Table 1 of the revised manuscript.
>     These results confirm that LMABO's strategy provides substantial value over a random policy, indicating that the LLM is extracting meaningful signals from the optimization state.
> 2. **Verifying the Decision Logic**
>     To understand how the LLM takes its decision, our original submission went beyond the text justifications and conducted a rigorous Information Sensitivity Analysis (Section 6, Appendix G).
>     By systematically perturbing individual state variables (e.g., remaining budget, current $f_{min}$, etc.) while keeping others fixed, we probed the link between the input state and the LLM's decision.
>     Our results (Tables 9-12) show consistent, logical behavior: for instance, reducing the remaining budget triggers a switch from explorative AFs (e.g., PES) to exploitative ones (e.g., PosMean). This empirical probing proves that the LLM's decisions are functionally dependent on the provided context and align with rational optimization principles.
> 3. **Validity of Textual Justifications**
>     Finally, regarding the validity of the generated justifications, in our original submission, we analyzed the semantic consistency of the LLM's output in Figure 2d (now Figure 3 in the revised manuscript).
>     We found a strong correlation between the selected acquisition function category and the keywords in the justification (e.g., terms like 'uncertainty' and 'exploration' appear predominantly when explorative AFs are chosen).
>     This suggests that the text justification is not a hallucination, but an accurate reflection of the model's internal decision logic.
>
> Along with other analyses in Section 6, we believe that these additional experiments provide strong evidence that the LLM's AF selections are both strategically grounded in the optimization state and logically consistent with how an expert would reason about the problem.
>
> **Regarding adding a new acquisition function**
> We acknowledge the reviewer's valid point regarding LMABO's reliance on the LLM's pre-trained knowledge.
> However, while we believe that adding brand-new AFs is possible in principle, it would not lead to meaningful improvements.
> Firstly, although LMABO leverages the LLM's pre-trained knowledge, it is not strictly bound by it.
> A brand-new AF can be described in the prompt (e.g., its mathematical definition, intended purpose, exploration-exploitation characteristics, etc.), and the LLM can reason about its potential utility in the current optimization context.
> On the other hand, adding brand-new AFs may not lead to significant performance gains.
> Our portfolio, consisting of 12 well-established AFs, already covers a broad spectrum of exploration-exploitation trade-offs and optimization strategies.
> These options are widely validated in the literature, providing a robust foundation for the LLM to exercise its strategic reasoning on a variety of optimization scenarios.
> Furthermore, by expanding the portfolio with unproven or niche AFs, we risk diluting the LLM's focus and potentially introducing noise into its decision-making process.
> In summary, while LMABO can incorporate new AFs through prompt engineering, we believe that the existing portfolio with proven AFs should be sufficient for effective optimization across diverse tasks.
>
> **Regarding exploiting prior knowledge**
> Thank you for this suggestion.
> We agree that integrating prior knowledge could potentially enhance LMABO's performance, especially in scenarios where such information is available.
> We have conducted additional experiments to test this hypothesis on both synthetic and hyperparameter optimization tasks, where we add prior knowledge about the objective function as initial context to the initial prompt (e.g., smoothness, periodicity, number of local minima, etc.).
> Results and discussions of these experiments have been added to the end of Section 5.3.
> In short, we find evidence that this additional context can further improve LMABO's performance, especially in phases where stagnation occurs.
> We have added details about this context in Appendix C.1 for reproducibility.
>
> **Regarding minor presentation issues**
> We have addressed all the minor presentation issues raised by the reviewer in the revised manuscript.

---

> > ### Comment · Reviewer_Sjvn · 2025-11-27
> >
> > Thank you for the detailed rebuttal and additional experiments.
> >
> > I acknowledge that LLM-based AF selection has its benefits (e.g., incorporating prior knowledge about the problem or budget constraints), but ultimately, LMABO proposes to address a black-box optimization problem (e.g., minimizing $f$) by introducing another layer of complexity that involves another black box (the LLM).
> >
> > Although empirical results are good (hence my positive score), the lack of formal understanding about the behavior of LLMs propagates to the behavior of LMABO. Therefore, there is no formal way to precisely describe why LMABO is performing well on the considered benchmarks, and to precisely understand what are its limitations. This prevents me from sending a stronger positive signal for this paper.
> >
> > Thus, I will keep my score as is.

---

> > > ### Author Response · Authors · 2025-11-28
> > >
> > > Thank you for highlighting the black-box nature of LLMs. We agree that, unlike traditional acquisition functions (e.g., UCB with its $\sqrt(\beta_t)$ regret bounds), an LLM-based strategist currently lacks a closed-form convergence proof.
> > >
> > > However, we respectfully argue that even in the absence of such a formal understanding in the form of theoretical convergence with the use of LLMs, we have established a precise **behavioral understanding** of why LMABO performs well. Our analysis in Section 6 shows that the LLM is not acting randomly, but rather as a **proxy for human expert intuition**. We have empirically verified that the model mimics specific, rational strategies that a human practitioner would adopt.
> > >
> > > Our Information Sensitivity Analysis (in Section 6) and Behavioral Analysis (in Figures 2 and 3) reveal that the LLM has successfully internalized the "best practices" of Bayesian Optimization from its pre-training data. It executes clear, explainable heuristics expected by an expert:
> > > - **Horizon-aware Exploitation**: Just as a human expert would "cash in" near the deadline, the LLM systematically shifts to exploitative AFs (e.g., PosMean, EI) when the remaining budget is low.
> > > - **Active Stagnation Breaking**: When a human observer sees the optimization stalling in a local optimum or failing to explore far from existing points, they switch to high-variance exploration. LMABO mimics this: our sensitivity analysis shows it switches to explorative functions (like Thompson Sampling) specifically when $f_{min}$ stagnates.
> > > - **Momentum Exploitation**: When a new optimum is found, the LLM sticks to improvement-based functions (e.g., EI) to refine the mode, mirroring the standard "exploit while promising" heuristic. In addition, if the improvement is small, it prioritizes LogEI instead of EI for better numerical stability, which is a common practice in BO.
> > >
> > > Therefore, we argue that we *can* describe why LMABO performs well: it succeeds because it approximates the decisions a human expert would make by reading the state of the optimization. We believe this **behavioral alignment with expert heuristics** serves as a robust proxy for formal understanding in the context of large language models.

---

### Official Review · Reviewer_KYRT · 2025-10-30

**Soundness:** 3
**Presentation:** 2
**Contribution:** 3
**Rating:** 6
**Confidence:** 3

**Summary:**

The paper introduces LMABO (LLM-Assisted Adaptive Bayesian Optimization), a novel framework that utilizes a pre-trained Large Language Model (LLM) as a zero-shot, online strategist to dynamically select the most suitable acquisition function (AF) at each iteration of the Bayesian Optimization (BO) process. LMABO addresses the limitation of existing adaptive methods by employing a structured state representation that translates the complete optimization context—including remaining budget, performance history, and surrogate model (GP) characteristics—into a text prompt for the LLM to reason over. Evaluated across 50 benchmark problems, LMABO demonstrated significant performance improvement over static, adaptive portfolio, and other LLM-based baselines (e.g., 55% lower total AUC than the best LLM-based method), confirming that its success is due to an emergent, robust, and state-aware policy for controlling the exploration-exploitation trade-off.

**Strengths:**

* The paper introduces a framework that successfully recasts the task of acquisition function (AF) selection in Bayesian Optimization (BO) as an in-context decision-making problem.   This approach leverages a pre-trained Large Language Model (LLM) to select the most appropriate AF at each optimization step based on its implicitly encoded knowledge of optimization principles.
* A primary technical contribution is the design of a structured state representation that translates the complex, multi-faceted numerical state of the BO process into a concise textual summary.   This summary includes crucial strategic information, such as the remaining optimization budget and insights from the surrogate model's GP lengthscales, overcoming the limitation of prior adaptive methods that relied only on narrow performance history.
* Extensive experiments across a diverse set of 50 benchmarks show that LMABO achieves a significant performance improvement over established static, adaptive portfolio, and other LLM-based baselines.   Analysis confirms that the framework’s effectiveness stems from an emergent, state-aware policy that executes active, dynamic switching between exploratory and exploitative AFs in response to real-time progress , demonstrating an agility beyond simple pre-defined heuristics.

**Weaknesses:**

* The LMABO algorithm requires calling the Large Language Model at every single optimization step. Although the financial cost per token is very low, the overall economic feasibility for extensive or complex problems using commercial LLM APIs requires more detailed analysis.
* The framework's high performance is fundamentally dependent on the sophisticated reasoning capabilities of the underlying LLM. The core implementation relies predominantly on Gemini-2.5 Flash. Ablation studies confirm this sensitivity, showing a noticeable drop in performance when using smaller open-source models, alongside a dramatic decrease in time efficiency. To ensure the framework's broad reliability and generalizability, its efficacy requires additional, comprehensive validation across a wider spectrum of diverse and competitive LLMs.

**Questions:**

* Could the authors provide a more detailed analysis of the economic costs associated with utilizing the LMABO framework?

* The core, high-performing LMABO implementation relies predominantly on Gemini-2.5 Flash.   Given that ablation studies demonstrate a noticeable performance drop when using smaller open-source models, could the authors verify the framework's broad reliability and efficacy by testing its performance and execution speed with other competitive closed-source LLMs (beyond Gemini-2.5 Flash) or high-capability open-source models from different architectures?

---

> ### Author Response · Authors · 2025-11-23
>
> We thank the reviewer for the thoughtful and constructive feedback. Below are our responses to the main comments.
>
> **Regarding economic feasibility of the LMABO framework**
> We believe that there is a misunderstanding about the cost of running LMABO.
> The reviewer mentions "Although the financial cost per token is very low", which seems to stem from our reported cost in Section 5.2 (now line 316 in the revised manuscript).
> However, what we meant was that the total cost of running LMABO for a full optimization run (i.e., 50 iterations) is only $\approx$\$0.01, and this is not the cost per token.
> Again, while the relative weight of the LLM cost compared to the function evaluation cost will vary by application, we believe that in many practical BO scenarios, the additional cost of LMABO will be negligible compared to the total cost of function evaluations.
>
> **Regarding the dependence on Gemini-2.5 Flash**
> We agree with the reviewer that evaluating LMABO with more diverse LLMs would strengthen our claims.
> We have added ablation studies with two additional LLMs: GPT-4o mini (for closed-source alternatives) and gpt-oss-120b (for high-capability open-source alternatives).
> Their performance can be found in Table 2, and their runtimes are reported in Table 5.
> Most importantly, we observe that both GPT-4o mini and gpt-oss-120b achieve similar performance compared to Gemini 2.5 Flash, demonstrating the reliability of LMABO across different LLMs and that it is not overly dependent on Gemini-2.5 Flash.
> Again, we thank the reviewer for this valuable suggestion.

---

### Official Review · Reviewer_HUN7 · 2025-10-31

**Soundness:** 3
**Presentation:** 1
**Contribution:** 2
**Rating:** 4
**Confidence:** 4

**Summary:**

This paper proposes LMABO, a zero‑shot strategy that delegates acquisition-function selection in Bayesian Optimization to Large Language Model. At each BO iteration, the model receives a state with: optimization state—process status (iteration count, remaining budget, dimensionality), performance history. This information is useful for the LLM to sample the best next acquisition function to be used.   Basically the method can be seen as an LLM doing a meta decision over an additional component of the acquisition process. The authors show strong result, which highlights the significance of the paper.

**Strengths:**

**Strong results.** Casting AF selection itself as an in‑context decision problem for an LLM shows that the meta decision of selecting the acquisition function is highly relevant for the whole acquisition process. The paper shows strong results.

**Weaknesses:**

**Novelty.** While the framing is neat, the algorithmic contribution largely reduces to a prompt + a state serialization + a portfolio choice. Many recent works have leveraged LLM inductive biases for decision selection or candidate generation in BO, so the conceptual step—“use an LLM to pick an AF given a textual state”—feels incremental without additional design elements.


**What paper should have.** The paper positions an LLM as the decision-maker for selecting the acquisition function, with the main contribution lying in the experiments. Two questions should be addressed: (1) Can the LLM reliably choose an appropriate acquisition function at each point in time? and (2) Do the LLM’s choices align with—or diverge from—established expectations, given that its reasoning comes from in-context learning? The paper convincingly demonstrates (1). It attempts to address (2), but the presentation could be stronger: the Background section does not give readers enough context to evaluate why the LLM makes the choices it does.

**Results.** The ablation studies are helpful but insufficient. The experimental setup around Figure 1 seems to be underspecified—it’s unclear whether the figure is illustrative or aggregates results across multiple runs. Results in Figure 2 show high variance, which makes it difficult to draw robust conclusions about the method’s behavior

**Questions:**

See weaknesses

---

> ### Author Response · Authors · 2025-11-23
>
> We thank the reviewer for the careful reading and constructive feedback. Below are our responses to the main comments.
>
> **Regarding novelty concern**
> Firstly, while we acknowledge that LLMs have been applied to BO in prior works, as detailed in our response to reviewer iyWi, LMABO differs structurally from existing works.
> While methods like FunBO operate as offline code generators (algorithm discovery) and LLAMBO/LLMP act as component replacements (surrogate modeling/candidate generation), LMABO establishes the "semantic controller" paradigm.
> We move the LLM out of the role of numerical regression or code generator and into the role of high-level strategic decision maker.
> By abstracting the optimization state into natural language, we leverage the LLM's native reasoning capabilities to orchestrate rigorous mathematical tools (standard AFs) rather than attempting to replace them.
> This allows for zero-shot, online adaptation that is structurally impossible for offline methods like FunBO.
>
> Furthermore, LMABO, besides the use of LLM, is the first adaptive BO work to empirically verify that integrating auxiliary state information (e.g., remaining budget, GP model characteristics, etc.) is an effective driver of adaptive performance.
> Our ablation studies (Table 2) confirm that this "rich state" synthesis is not merely a data dump, but a critical scientific contribution: it proves that exposing these latent optimization states is a prerequisite for effective agentic control.
>
> Finally, regarding the simplicity of the proposed method, we adhere to the principle of parsimony.
> In optimization literature, complexity often brings about brittleness.
> We intentionally prioritized a streamlined architecture to demonstrate that a well-designed semantic interface alone is sufficient to solve the adaptive BO problem.
> The validity of this approach is confirmed by our results in which our ``simple" method significantly outperforms sophisticated, computationally intensive baselines.
> We argue that achieving state-of-the-art performance with an elegant, zero-shot framework highlights the robustness of the semantic controller paradigm, whereas adding further design elements would likely introduce unnecessary complexity without corresponding gains.
>
> **Regarding a better explanation for LLM's choices**
> We thank the reviewer for this insight. We first have amended the Background section (Section 2) to better explain the exploration-exploitation spectrum of AFs and how different AFs can be categorized into explorative vs. exploitative groups.
> We believe that this additional context, along with the existing information in this section, will help readers better understand the factors to be considered by an effective AF selection strategy (not just LMABO but also other adaptive portfolio methods).
> Secondly, we would like to add that Section 6 of the original submission already contains an in-depth analysis of LMABO's AF selection behavior, showing that it aligns closely with established best practices in BO.
> For instance, we showed that the LLM "heavily exploits with low remaining budget", which aligns with the standard expectation that exploration should decrease as the optimization horizon shortens.
> Another pattern is that LMABO "decisively switches back to exploratory functions to escape stagnation", consistent with the practice of preventing the optimization from getting trapped in local optima.
> To highlight these alignments more clearly, we have added comments to the revised manuscript (in lines 62-64, the beginning of Section 6, and the first paragraph in Section 7) to connect LMABO's observed behaviors with established BO principles.
>
> **Regarding insufficient results**
> About Figure 1 (in the original submission), we have added the clarification that the results are aggregated across all runs on all 50 benchmark problems at the beginning of Section 6 and the caption of Figure 2 in the revised manuscript.
> About Figure 2 (in the original submission), please check our response to reviewer iyWi regarding the improved presentation.
> In short, we have added meta-strategies to the main experiments, expanded their benchmark set to all 50 problems, and applied the same statistical tests as other baselines to better illustrate the performance differences.

---

### Official Review · Reviewer_iyWi · 2025-10-31

**Soundness:** 2
**Presentation:** 1
**Contribution:** 2
**Rating:** 4
**Confidence:** 4

**Summary:**

The paper studies meta–decision making in Bayesian Optimization. Specifically, how to select the acquisition function (AF) for a given task/budget—and proposes using a Large Language Model (LLM) as a source of prior knowledge to guide that choice. Concretely, the method extracts task context (e.g., problem description, dataset traits, search‑space information, etc) and queries an LLM to recommend an AF (e.g., EI, UCB). The authors evaluate this idea across a broad set of AutoML/HPO problems and report that making the right meta decision (AF selection) substantially affects BO performance.

**Strengths:**

The work tackles a meta design choice in BO/AutoML—which AF to use—rather than only tuning model hyperparameters. In practice this could translate in a more robust “out‑of‑the‑box” optimizer that requires less trial to find the optima.

The paper reports extensive experiments indicating that the meta decision is relevant. It is important to have flexibility in what AF to choose along the acquisition process.

**Weaknesses:**

Presentation. I think the presentation and how the paper is written is not good in general. To give some examples. 1) Some figures are hard to read or to use for significance judgments. In Figure 2, the variance of the result is  wide, making it difficult to visually assess the significance of the results. 2) Table 2 uses cryptic labels (“LMABO‑AB1…AB4”); these should be replaced or augmented with descriptive names.
Novelty. The idea of leveraging LLM “knowledge” to steer BO is not brand‑new (e.g., LLAMBO, LLMP, FunBO; Sec. 3), although applying it specifically to the meta decision of AF selection is useful as the authors have shown empirically.
Ablation. The paper includes several ablation studies, which is valuable. However, this should also be examined at a more global level to demonstrate that the observed patterns consistently repeat across multiple runs and different experiments.

**Questions:**

See above

---

> ### Author Response · Authors · 2025-11-23
>
> We appreciate the reviewer's positive feedback and constructive comments. Below are our responses to the main points raised.
>
> **Regarding presentation**
> We thank the reviewer for the suggestions on improving the presentation of our work.
> Many of these shortcomings were due to space constraints in the original submission, and we have addressed them in the revised manuscript.
> For Figure 2 in the original version, we agree that the min-max shadings (not variance) were too wide to assess the significance of the performance differences.
> In the revised manuscript, we have added the meta-strategies in this plot to the main experiment (with an added introduction in Section 5.1), expanded the benchmark set for these meta-strategies to all 50 problems instead of just 4, and applied the same statistical tests as other baselines to better illustrate the performance differences.
> Given the new results (now presented in Table 1 of the revised manuscript), we observe that our claim about LMABO outperforming meta-strategies still holds strongly.
> The discussion about these meta-strategies has also been moved to the end of Section 5.2.
> In addition, we also remove the cryptic labels in Table 2 with more descriptive names for clarity.
>
> **Regarding novelty**
> We appreciate the reviewer acknowledging the utility of applying LLMs to AF selection.
> However, we believe there is a fundamental distinction in how the LLM is utilized in LMABO compared to baselines like LLAMBO and LLMP:
> 1. **Semantic Reasoning vs. Numerical Processing (Architectural Alignment)**
> A critical distinction lies in the modality of the task.
> Methods like LLMP and LLAMBO typically treat the LLM as a numerical processing unit, tasking it with acting as a surrogate model (regression) or directly sampling candidate points.
> While innovative, this approach requires the LLM to operate outside its native strengths, often leading to challenges with numerical precision and tokenizer-induced hallucinations (Shao et al. 2025).
>
>     LMABO differentiates itself by abstracting the numerical state into a semantic representation.
>     Instead of feeding raw arrays to the LLM to perform arithmetic, we serialize the optimization dynamics into high-level concepts (e.g., "process is stagnating," "lengthscales are varying," "budget is tight").
>     This aligns the BO task with the LLM's pre-training objective: text-based reasoning and logical inference.
>     We argue that this is a more natural and robust way to integrate LLMs into the optimization loop: we keep the precise calculations (GPs, acquisition values) in the mathematical engine, and use the LLM exclusively as the semantic strategist to control the trade-off.
>
> 2. **Distinction from FunBO (Offline Code Discovery vs. Online Zero-Shot Strategy)**
> LMABO is an online, zero-shot strategist, not attempting to invent new code or train a policy.
> It accepts that no single AF is optimal and dynamically orchestrates a portfolio of existing tools based on real-time state.
> This distinction remedies a key limitation explicitly noted by the FunBO authors: the 'computational overhead associated with running a full BO loop'.
> Because FunBO requires expensive offline evolutionary search to find a policy, it lacks the immediate, zero-shot adaptability of LMABO, which steers the optimization 'on the fly' without any pre-training overhead.
> Thus, FunBO is Algorithm Discovery, whereas LMABO is Adaptive Control.
>
> These distinctions highlight a fundamentally different philosophy in how LLMs are integrated into BO in LMABO, focusing on leveraging their core strengths in semantic reasoning for dynamic strategy selection rather than numerical computation or offline code generation.
> In addition, Bayesian optimization is inherently a composite framework of multiple components, and the integration of LLMs into BO can be for various purposes.
> Thus, while the idea of using LLMs in BO has been explored, we believe our specific formulation and execution of LMABO represents a novel and meaningful contribution to the field that complements existing works, rather than simply rehashing prior ideas.
> Having said that, we have added more clarifications to Section 3 to better highlight these distinctions.
>
> Shao, J., Lu, Y., \& Yang, J. (2025). Benford's Curse: Tracing Digit Bias to Numerical Hallucination in LLMs. The Thirty-Ninth Annual Conference on Neural Information Processing Systems.
>
> **Regarding more global examination for the observed patterns**
> In the original submission, we forgot to describe in detail that the observed patterns in Section 6 are aggregated across all runs on all problems.
> We have added this clarification at the beginning of Section 6 and the caption of Figure 2 in the revised manuscript.

---

### Author Response · Authors · 2025-11-23

We would like to express our gratitude to all reviewers for their thorough evaluations and insightful feedback.
The list of major changes made in the revised manuscript includes:

* Replaced Figure 2 in the original submission by adding the simple meta-strategies to the main experiments in Table 1, expanding their benchmark set to all 50 problems, and applying the same statistical tests as other baselines to better illustrate the performance differences. The corresponding discussion has also been moved to the end of Section 5.2.
* Added random baselines as simple meta-strategies to Table 1 to validate that LMABO's performance gain is due to strategic decision-making rather than random chance.
* Added ablation studies with two additional LLMs (GPT-4o mini and gpt-oss-120b) to Table 2 to demonstrate the reliability of LMABO across different LLMs.
* Added an experiment and discussed results on exploiting prior knowledge about the objective function in Section 5.3 (Figure 1 in the revised manuscript) to show that LMABO can further benefit from such information when available.

Note that by adding more methods to the main experiments, the relative rankings of some baselines have slightly changed compared to the original submission, but all conclusions still hold strongly.

Besides these major changes, we have also made numerous minor revisions throughout the manuscript to improve clarity, presentation, and address specific reviewer comments:
* Added clarifications about the aggregation of results in Section 6 and the caption of Figure 2.
* Added comments to connect LMABO's observed behaviors with established BO principles in lines 62-64, the beginning of Section 6, and the beginning of Section 7.
* Added more introduction about the exploration-exploitation spectrum of AFs in Section 2 to help readers better understand the factors to be considered by an effective AF selection strategy.
* Added comments to discuss the differences between the approach of LMABO and prior LLM-based BO works in Section 3.
* Modified the explanation of the cost of running LMABO in Section 5.2 to avoid misunderstanding.
* Moved Figure 2d in the original submission to Figure 3 in the revised manuscript for better presentation.
* Replaced cryptic labels in Table 2 with more descriptive names for clarity.

Again, we sincerely thank the reviewers for their valuable feedback, which has strengthened the rigor and completeness of our manuscript.

---

> ### Author Response · Authors · 2025-12-02
> **Summary of the discussion**
>
> Due to the recent data leakage on OpenReview, we are unable to engage in further discussions with the reviewers, having received only one response so far.
> We deeply regret missing the opportunity to fully explore the nuances of the feedback, but we have earnestly addressed all the comments raised in our responses and the revised manuscript.
> To help the AC make an informed decision, we have summarized the main points of discussion below:
>
> 1. **Novelty: The "Semantic Controller" Paradigm** (Reviewers iyWi and HUN7)
> Reviewers asked for clarification on how LMABO differs from prior LLM-based BO works.
> We clarified that LMABO introduces a distinct semantic controller paradigm.
> Unlike offline code generators (FunBO) or component replacements that attempt numerical regression (LLAMBO/LLMP), LMABO leverages LLM's native strength - semantic reasoning - to dynamically orchestrate a portfolio of established acquisition functions, enabling zero-shot, online adaptation, and is also the first work to consider rich auxiliary state information for adaptive control in BO.
>
> 2. **On how the LLM makes decisions** (Reviewers HUN7 and Sjvn)
> Regarding the decision-making process of the LLM, we hypothesize that the LLMs succeed in this task by mimicking the heuristic strategies of human experts found in their pre-training data.
> We provided robust empirical evidence in Section 6 to validate this mimicry by showing alignment between LMABO's behavior and established BO practices, for example:
> - Horizon-awareness: The LLM systematically shifts to exploitation when the budget is low.
> - Stagnation breaking: It actively switches to high-variance exploration when progress stalls.
> - Momentum exploitation: It utilizes improvement-focused AFs immediately after finding new optima.
>
> We argued that this behavioral alignment with rational expert strategies indicates that the LLM is not making arbitrary choices but is grounded in meaningful optimization principles, explaining why LMABO performs effectively.
>
> 3. **Robustness across LLMs** (Reviewer KYRT)
> We validated that LMABO's performance is robust across different LLMs by adding ablation studies with GPT-4o mini and gpt-oss-120b.
> Both LLMs achieved similar performance compared to Gemini 2.5 Flash, demonstrating that LMABO is not overly dependent on a specific LLM.
>
> Besides these main points, we have also addressed all other comments and required clarifications raised by the reviewers in our responses and the revised manuscript.
> We believe that the revised manuscript is significantly strengthened and provides a more comprehensive framework that not only demonstrates superior empirical performance but also provides a clear, verifiable explanation for the mechanism behind this success.

---

### Meta-Review · Area_Chair_wCQu · 2026-01-09

**Summary:**

This paper proposes LMABO, a zero-shot method that delegates acquisition-function selection in Bayesian Optimization (BO) to a Large Language Model (LLM). At each optimization step, the LLM receives a structured description of the current optimization state and historical performance metrics. Based on this context, the LLM selects the acquisition function to be applied in the subsequent iteration. At its core, LMABO augments BO with an LLM-based meta-controller over acquisition-function choice.

The main strengths of this work:
- This paper offers a new framing for LLM to take part in the BO loop through the meta decision of acquisition function.
- One main technical contribution is the design of a structured state representation that translates the multi-faceted numerical state of the BO process into an informative textual summary.
- The paper provides extensive experiments on a variety of benchmark tasks, with strong results showing that the meta decision is critical and it is important to have flexibility in the choice of AF during the acquisition process.

The reviewers also raised the following main concerns:

(1) Novelty (Reviewers: HUN7, iyWi):

Multiple reviewers note that the method largely amounts to prompt design and portfolio selection, which overlaps with existing approaches that also use LLM inductive biases for decision or candidate selection in BO (e.g., LLAMBO, LLMP, and FunBO), despite that its specific application to the meta decision of AF selection appears useful based on the experiments.



(2) Validity and interpretability of LLM decisions (Reviewers: HUN7, Sjvn)

Reviewers also expressed concern that although the LLM provides justifications, there is no guarantee that these explanations reflect the true decision process or align with the established BO principles. Additionally, reliance on the LLM’s implicit prior knowledge raises questions about whether LMABO can adapt to newly introduced acquisition functions.


(3) Cost and sensitivity to the choice of LLM (Reviewers: KYRT, iyWi)

Reviewers noted that querying an LLM at every BO step can be very costly, especially for large-scale or long-horizon BO problems, which are not sufficiently analyzed. Moreover, performance appears tightly coupled to the capabilities of Gemini-2.5 Flash, with degradation when using smaller or open-source models.

(4) Presentation (Reviewers: HUN7, Sjvn, iyWi)

Several reviewers highlighted that key figures are difficult to interpret due to the large variance, unclear aggregation across runs, or hard-to-read captions. While ablation studies are appreciated, they are considered incomplete without demonstrating consistent patterns.

**Reviewer Concerns:**

After the rebuttal, the first two main concerns are to some extent alleviated, and the concerns (3) and (4) have been fully addressed. Specifically:

As for (1), in the rebuttal, the authors clarified that LMABO treats LLM as a semantic controller for online AF selection, in contrast to LLM as a numerical regressor in LLAMBO and LLMP and LLM as a tool for algorithm discovery in FunBO.

Regarding (2), during the rebuttal, the results in Section 6 were highlighted and augmented to interpret the underlying decision making of LLMs. This alleviated the concerns on the interpretability of LLM decision.
On the other hand, the authors also acknowledged that adding a brand-new AF to the portfolio for improvement can be a challenge given the pre-trained knowledge of LLMs.

Regarding (3), the rebuttal reported the token consumption per BO run, which appears mild. Moreover, the authors added new results on various other LLMs beyond Gemini-2.5 Flash. This alleviated the concern on the dependency on the specific LLM.

The revision addressed the aforementioned figure / table / caption issues and improved the presentation of the experiments.

**Reviewer Scores:**

This submission is rather borderline with mixed reviews: iyWi: 4 / HUN7: 4 / KYRT: 6 / Sjvn: 6.

Upon a careful read of the paper, the reviews, and the rebuttal, the major concerns mentioned above appear mostly answered despite the high-level resemblance to the existing LLM-based methods for BO.

In summary, this paper offers an interesting new way to integrate BO with LLMs and can complement the existing LLM-based research in this area. The experimental results are promising and offer some interpretability into the LLM decisions. Overall the strengths of the paper outweigh its remaining limitations, and I therefore recommend acceptance.

---

### Decision · Program_Chairs · 2026-01-26

Accept (Poster)